# Bridging Local–Global Dissonance: Learning from Compressive Measurements for Hyperspectral Reconstruction

**Xian-Hua Han** [1]

## Abstract

Reconstructing hyperspectral images from compressive measurements is challenging due to a fundamental mismatch between locally reliable observations and globally entangled structures induced by spectral dispersion. This study formalizes this issue as a local–global dissonance in representation learning for CASSI systems. To resolve it, we propose a Hierarchical Scale-Reconciling Architecture (HSRA) that enforces local sufficiency and global consistency in a principled, scale-aware manner. HSRA combines multi-kernel token mixing, latent window interactions, and hierarchical multi-granularity spatially shifted attention to progressively reconcile physical constraints across scales. Embedded into a deep unfolding framework as a physically grounded learned prior, Extensive experiments on benchmarks demonstrate that HSRA achieves consistent and significant improvements over state-of-the-art methods. Code will be publicly available at: https://github.com/hanxhua1216/HSRA.

## 1. Introduction

Hyperspectral (HS) imaging represents scenes as spatial–spectral three-dimensional data cubes, capturing dense spectral distributions that go far beyond conventional RGB imagery and enabling a wide range of applications, including remote sensing (Borengasser et al., 2007; Melgani & Bruzzone, 2004; Solomon & Rock, 1985; Ding et al., 2024), medical imaging (Zhi et al., 2007; Fei, 2019), and environmental monitoring (Wright et al., 2019). Conventional HS acquisition systems obtain this dimensionality through sequential scanning across spectral bands (Ma et al., 2021),

which inherently limits temporal resolution. To overcome this constraint, Coded Aperture Snapshot Spectral Imaging (CASSI) (Gehm et al., 2007; Arce et al., 2013; Wang et al., 2015) leverages compressed sensing (CS) principles (Ma et al., 2021; Cao et al., 2016; Llull et al., 2013) to optically encode a full 3D HS cube into a single 2D snapshot. While this multiplexing strategy enables snapshot acquisition, it fundamentally alters the structure of the measurements: pixel-wise coded apertures preserve locally reliable spatial–spectral statistics, whereas spectral dispersion induces spatial–spectral entanglement that distorts long-range dependencies. As a result, the resulting measurements impose incompatible local and global constraints on the latent HS signal, giving rise to a pronounced local–global dissonance that renders reconstruction from incomplete observations severely ill-posed. Despite substantial progress in both model-based (Dias & Figueiredo, 2007; Liu et al., 2018) and learning-based (Ma et al., 2019; Hu et al., 2022; Miao et al., 2019) reconstruction methods, effectively reconciling locally faithful but globally entangled measurements remains a central challenge in modern HSI restoration.

Early HS image reconstruction methods address this inverse problem by imposing hand-crafted priors, such as total variation (Eason & Andrews, 2014; Dias & Figueiredo, 2007), low-rank structure (Liu et al., 2018), and sparsity (Zhang et al., 2018; Figueiredo et al., 2017). Although these approaches are mathematically well-founded, they often suffer from high computational complexity and limited adaptability to diverse scenes. More recently, deep learning has emerged as a dominant paradigm, encompassing both end-to-end regression models (Ma et al., 2019; Hu et al., 2022; Cai et al., 2022b) and deep unfolding methods (DUMs) (Yuan et al., 2020; Wang et al., 2022a; Cai et al., 2022c; Li et al., 2023). Convolutional neural networks (CNNs) (Miao et al., 2019; Wang et al., 2019) have been widely adopted within these frameworks due to their strong locality bias, yet this same bias restricts their ability to recover long-range spatial–spectral dependencies severely distorted by the CASSI measurement process. Conversely, Vision Transformers (ViTs) (Dosovitskiy et al., 2020; Liu et al., 2021) and spectral-wise attention models (Cai et al., 2022b; Takabe et al., 2024; Dong et al., 2023) offer powerful global modeling capabilities, but their permutation-equivariance

[1] Graduate School of Artificial Intelligence and Science, University of Rikkyo, Tokyo, Japan. Correspondence to: Xian-Hua Han <hanxhua@rikkyo.ac.jp>.

*Proceedings of the 43rd International Conference on Machine Learning*, Seoul, South Korea. PMLR 306, 2026. Copyright 2026 by the author(s).

self-attention mechanisms implicitly assume globally comparable token semantics, an assumption violated by entangled compressive measurements. Under noise and extreme undersampling, vanilla Transformers tend to amplify spurious long-range correlations and propagate local artifacts globally, while incurring prohibitive computational costs for high-resolution HS image. These limitations underscore the need for a principled and efficient architecture that aligns learned representations with the physical measurement process, reconciling local structural fidelity with global contextual coherence.

In this study, we propose a Hierarchical Scale-Reconciling Architecture (HSRA) explicitly designed to reconcile the local–global dissonance induced by compressive HS measurements. Rather than applying uniform global modeling, our architecture decomposes the reconstruction process into complementary stages that align with the physical and statistical structure of CASSI observations. At the initial high-resolution encoding stage, we introduce a structured interface composed of a Multi-Kernel Perception Token Mixer (MK-PTM) and a Latent Window Interaction Module (LWIM). MK-PTM injects strong local inductive biases to extract locally reliable spatial–spectral statistics preserved by pixel-wise coded apertures, while LWIM aggregates window-level representations into a compact latent space where global interactions can be modeled efficiently and robustly. This separation of local aggregation and global interaction prevents the premature propagation of measurement-induced artifacts that commonly arise in vanilla self-attention. To progressively enforce global coherence, we employ Hierarchical Multi-Granularity Spatially-Shifted Attention (HM-SSA) across subsequent encoder–decoder stages, systematically expanding attention granularity from fine-scale local regions to coarser contextual partitions. The coarse-to-fine integration enables consistent alignment of long-range dependencies without sacrificing local structural fidelity.

The proposed HSRA is evaluated under two complementary learning paradigms. First, we consider an end-to-end (E2E) reconstruction setting, where three model variants of increasing capacity are trained to directly map compressive measurements to HS reconstructions, highlighting the representation learning capability of the proposed design. Second, we embed the same architecture into a deep unfolding framework, in which the HSRA serves as a physically consistent learned prior constrained by the CASSI forward model. This dual evaluation allows us to disentangle the benefits of architectural inductive bias from those of physics-guided optimization and to demonstrate robustness under noise and severe undersampling.

The main contributions of this work can be summarized as follows:

- We formalize local–global dissonance in compressive HS imaging, revealing how locally reliable but globally entangled measurements fundamentally limit existing CNN- and Transformer-based reconstruction methods.

- We propose a structured high-resolution encoding interface that combines Multi-Kernel Perception Token Mixing and Latent Window Interaction to ensure local sufficiency while enabling stable global reasoning.

- We introduce Hierarchical Multi-Granularity Spatially-Shifted Attention (HM-SSA), which progressively enforces global spectral–spatial coherence across scales within a U-Net hierarchy.

- We integrate the proposed architecture into a physics-consistent deep unfolding framework, yielding a learned regularizer aligned with the CASSI forward model and achieving improved accuracy and robustness under noise.

## 2. Related Work

### 2.1. Coded Aperture Snapshot Spectral Imaging

Coded Aperture Snapshot Spectral Imaging (CASSI) is a representative architecture for snapshot compressive HS imaging, enabling the acquisition of high-dimensional spectral information from a single two-dimensional measurement (Arce et al., 2013; Wang et al., 2015). By avoiding spatial or spectral scanning, CASSI significantly improves acquisition efficiency and hardware compactness, but introduces intrinsic challenges for faithful hyperspectral reconstruction.

Let the HS image be denoted as $\mathbf{X} \in \mathbb{R}^{H \times W \times B}$. The CASSI imaging process begins with local spatial modulation using a coded aperture $\mathbf{M} \in \mathbb{R}^{H \times W}$, producing band-wise modulated images

$$\mathbf{X}'_b = \mathbf{X}(:,:,b) \odot \mathbf{M}, \tag{1}$$

where $\odot$ denotes the Hadamard product. This operation preserves fine-grained local spatial statistics through pixel-wise constraints. Subsequently, a dispersive element shifts each modulated spectral band according to its wavelength, forming a spectrally sheared volume $\mathbf{X}''$,

$$\mathbf{X}''(h, w + d(\lambda_b - \lambda_c), b) = \mathbf{X}'(h, w, b), \tag{2}$$

where $d$ denotes the dispersion step and $\lambda_c$ refers to the central or reference wavelength. Finally, the detector integrates the sheared volume along the spectral dimension to produce the snapshot measurement

$$\mathbf{Y} = \sum_{b=1}^{B} \mathbf{X}''(:,:,b) + \mathbf{N}, \tag{3}$$

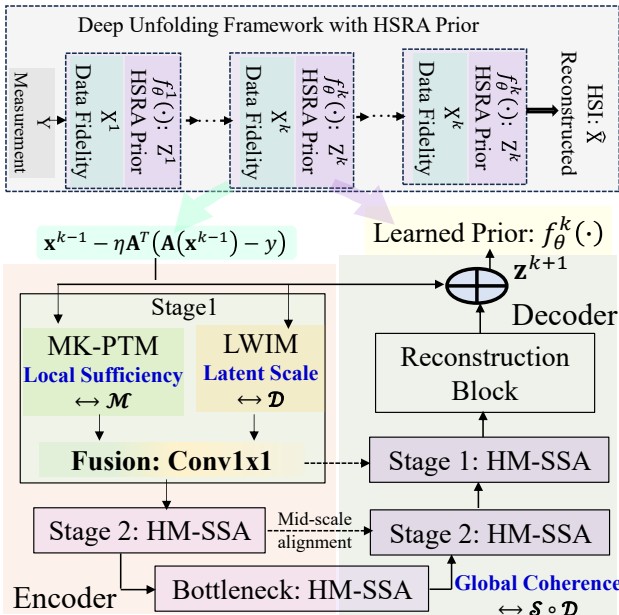

*Figure 1.* Overview of the proposed HSRA framework, highlighting the deep unfolding optimization process and hierarchical local–global dissonance-aware representation learning for HS image reconstruction.

with **N** representing measurement noise. From a representation learning viewpoint, spectral dispersion and integration introduce strong spatial–spectral entanglement in CASSI measurements. Although coded apertures preserve locally reliable constraints, the subsequent integration couples distant spatial and spectral components, yielding incompatible local and global requirements under severe compression. This intrinsic local–global dissonance motivates reconstruction priors that explicitly reconcile structural information across scales.

### 2.2. Evolution of Reconstruction Architectures

The evolution of learning-based HS reconstruction architectures reflects a gradual shift from strictly local representations toward increasingly global modeling, driven by the need to recover long-range spatial–spectral dependencies disrupted by snapshot sensing. Early CNN-based methods (Miao et al., 2019; Meng et al., 2020b) exploit local correlations through convolutional kernels and thus align well with the pixel-wise modulation imposed by coded apertures. Representative approaches demonstrate strong local fidelity and robustness to noise, but their inherently limited receptive fields restrict the recovery of non-local structures induced by spectral dispersion and integration, leading to blurred global consistency under severe compression (Meng et al., 2020b; Yorimoto & Han, 2021; Kohei & Han, 2020). To overcome these limitations, recent works have adopted Transformer-based architectures to explicitly model long-range dependencies (Cai et al., 2022b;a;

Takabe et al., 2024) . By leveraging self-attention mechanisms (Li et al., 2023; Zhang et al., 2024b), these methods significantly improve global coherence and spectral consistency. However, global attention is often introduced at early stages on high-resolution features, which can obscure fine-grained local statistics governed by the sensing mask and amplify measurement noise. In addition, the quadratic complexity of self-attention poses scalability challenges for high-resolution hyperspectral data.

Hybrid CNN–Transformer architectures attempt to combine local inductive biases with global context modeling. While effective in fully observed HS analysis tasks (Guo et al., 2024; Arshad et al., 2024; 2025; Sun et al., 2025), such designs are less suited to compressive imaging, where measurements are incomplete and globally entangled by the CASSI forward model. Without explicitly modeling the hierarchy of physical constraints, local and global representations (Wang et al., 2022b), remain loosely coupled, leaving the fundamental local–global dissonance unresolved.

### 2.3. Deep Unfolding and Optimization-Inspired Priors

Deep unfolding frameworks (Yuan et al., 2020; Zheng et al., 2021; Meng et al., 2020a; Ma et al., 2019; Wang et al., 2022a) reformulate iterative reconstruction algorithms as trainable networks, embedding the CASSI forward operator to preserve physical interpretability while learning data-adaptive priors. Compared with classical handcrafted regularizers, unfolding-based models (Cai et al., 2022c; Li et al., 2023; Qin et al., 2025; Hu et al., 2024; Zhang et al., 2024a; Han, 2025) provide greater expressive power to capture complex spatial–spectral correlations. Plug-and-Play approaches further replace analytical proximal operators with learned denoisers, improving flexibility and reconstruction quality. Nevertheless, most existing learned priors are trained as generic denoisers and remain weakly aligned with the sensing physics. As a result, their gradients may conflict with data-fidelity updates, particularly under severe undersampling and noise. This mismatch limits their ability to reconcile locally reliable coded observations with globally coherent reconstructions, underscoring the need for physically grounded, scale-aware priors that explicitly reflect the hierarchical structure of the CASSI measurement process.

## 3. Methodology

### 3.1. Problem Formulation and Local-Global Dissonance

Given the CASSI forward model, the compressive measurement process can be compactly expressed as a linear operator $\mathcal{A} : \mathbb{R}^{H \times W \times B} \to \mathbb{R}^{H \times (W + d(B-1))}$,

$$\mathbf{Y} = \mathcal{A}(\mathbf{X}) + \mathbf{N}, \qquad (4)$$

where $\mathcal{A} = \mathcal{S} \circ \mathcal{D} \circ \mathcal{M}$ denotes the composite sensing operator consisting of coded aperture modulation $\mathcal{M}$, spectral dispersion $\mathcal{D}$, and spectral integration $\mathcal{S}$. Reconstructing the latent HS datacube $\mathbf{X}$ from the compressed snapshot $\mathbf{Y}$ is a severely ill-posed inverse problem, commonly formulated as

$$\hat{\mathbf{X}} = \arg\min_{\mathbf{X}} \frac{1}{2}\|\mathcal{A}(\mathbf{X}) - \mathbf{Y}\|_2^2 + \lambda\mathcal{R}(\mathbf{X}), \qquad (5)$$

where $\mathcal{R}(\cdot)$ denotes a regularization functional encoding prior knowledge of the HS manifold.

To solve Eq. (5) while preserving physical interpretability, deep unfolding model adopts the Half Quadratic Splitting (HQS) strategy. By introducing an auxiliary variable $\mathbf{Z}$, the optimization is decomposed into two alternating sub-problems:

$$\begin{cases} \mathbf{X}^{k+1} = \arg\min_{\mathbf{X}} \|\mathcal{A}(\mathbf{X}) - \mathbf{Y}\|_2^2 + \mu\|\mathbf{X} - \mathbf{Z}^k\|_2^2, \\ \mathbf{Z}^{k+1} = \arg\min_{\mathbf{Z}} \frac{\mu}{2}\|\mathbf{Z} - \mathbf{X}^{k+1}\|_2^2 + \lambda\mathcal{R}(\mathbf{Z}), \end{cases}$$
$$\qquad (6)$$

where $\mu$ is a penalty parameter. The $\mathbf{X}$-update enforces data fidelity via a physics-driven proximal operator, while the $\mathbf{Z}$-update is implemented as a learnable neural prior,

$$\mathbf{Z}^{k+1} = f_{\boldsymbol{\theta}}^{k+1}(\mathbf{X}^{k+1}), \qquad (7)$$

with $f_{\boldsymbol{\theta}}^{k+1}(\cdot)$ denoting the network at the $(k+1)$-th stage.

The core challenge of CASSI reconstruction originates from the structural mismatch between the sensing operator $\mathcal{A}$ and conventional neural priors, which we formalize as Local–Global Dissonance. This arises from two competing requirements on the regularizer $\mathcal{R}$: 1) **Local Sufficiency vs. Undersampling.** The coded aperture modulation $\mathcal{M}$ enforces pixel-wise, high-frequency constraints, requiring strong local inductive biases to recover masked structures. However, due to severe undersampling, the forward operator $\mathcal{A}$ is locally ill-conditioned: reliable information at a single spatial location is dispersed across the detector plane. Consequently, purely local priors (e.g., convolutional models) lack sufficient context to disambiguate signal from noise. 2) **Global Consistency vs. Local Boundaries.** Spectral dispersion and integration ($\mathcal{D}$, $\mathcal{S}$) introduce long-range spatial–spectral coupling, making global structural constraints essential for disentangling spectral overlap. Yet, global operators such as standard self-attention often disregard the sharp, spatially varying boundaries imposed by the physical mask $\mathcal{M}$, leading to oversmoothing or spectral leakage.

This incompatibility is explicitly reflected in the unfolding dynamics through the data-fidelity gradient

$$\nabla_{\mathbf{X}}\mathcal{L}_{\text{fidelity}} = \mathcal{A}^{\top}(\mathcal{A}(\mathbf{X}) - \mathbf{Y}), \qquad (8)$$

where the back-projection $\mathcal{A}^{\top}$ propagates local measurement artifacts globally. As a result, a uniform prior, whether

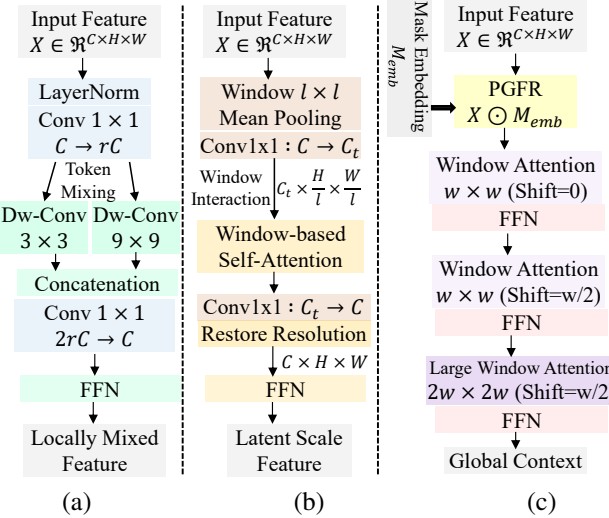

*Figure 2.* The detailed structures of (a) Multi-Kernel Perception Token Mixing (MK-PTM), (b) Latent Window Interaction Module (LWIM), and (c) Hierarchical Multi-Granularity Spatially-Shifted Attention (HM-SSA).

purely local or purely global, cannot satisfy both constraints simultaneously, motivating a physically grounded, multi-scale architecture that reconciles local sufficiency with global coherence.

## 3.2. Hierarchical Scale-Reconciling Architecture

To construct a hierarchical and physically grounded learned prior, we design $f_{\boldsymbol{\theta}}$ as a multi-scale reconciliation operator that systematically resolves the inverse problem by mapping hardware-level degradations to high-level HS manifolds. Rather than treating reconstruction as a uniform feature transformation, we explicitly decompose regularization into three interacting granularities, each corresponding to a distinct operator in the CASSI forward model and arranged within a UNet-like hierarchical architecture (Fig. 1).

Specifically, the learned prior integrates: (i) *local pixel-level sufficiency* via Multi-Kernel Perception Token Mixing (MK-PTM), (ii) *compact mid-range interaction* through a Latent Window Interaction Module (LWIM), and (iii) *global structural coherence* enforced by Hierarchical Multi-Granularity Spatially-Shifted Attention (HM-SSA). These components operate in a complementary and cooperative manner, forming a hierarchical prior that mirrors the composite structure of the CASSI forward operator $\mathcal{A} = \mathcal{S} \circ \mathcal{D} \circ \mathcal{M}$. In particular, local token mixing aligns with coded mask modulation $\mathcal{M}$, latent window interaction captures dispersion-induced coupling $\mathcal{D}$, and hierarchical attention models the global spectral integration imposed by spectral summation $\mathcal{S}$. This design ensures that the learned prior remains both physically consistent and hierarchically expressive, enabling effective reconciliation of local observations with global hyperspec-

tral structure.

**Multi-Kernel Perception Token Mixing for Local Sufficiency:** At the highest spatial resolution of the encoder, reconstruction is dominated by high-frequency discontinuities introduced by the coded aperture $\mathcal{M}$. To enforce local sufficiency at this stage, we employ a Multi-Kernel Perception Token Mixing (MK-PTM, illustrated in Fig. 2(a)) pathway that injects structured, mask-aligned inductive bias before any global interaction is introduced. Let $\mathbf{X} \in \mathbb{R}^{C \times H \times W}$ denote the input feature tensor at unfolding iteration $k$, corresponding to the back-projected estimate obtained via the adjoint operator. To mitigate the spatially heterogeneous noise amplification introduced by $\mathcal{A}^{\top}$ and to stabilize the subsequent optimization, the input is first processed by a spatially-aware Layer Normalization. A point-wise $1 \times 1$ convolution is subsequently applied to adjust the channel dimensionality to $rC$ with $r \leq 1$, yielding

$$\mathbf{X}r = \sigma\left(W_{1\times1}\mathrm{LN}(\mathbf{X})\right), \qquad (9)$$

where $\sigma(\cdot)$ represents the LeakyReLU activation function.

To encode scale-diverse local contextual information, the transformed features $\mathbf{X}_r$ are forwarded through parallel depth-wise convolutional branches with heterogeneous receptive fields,

$$\mathbf{P}_3 = \mathrm{DWC}_{3\times3}(\mathbf{X}_r), \qquad \mathbf{P}_9 = \mathrm{DWC}_{9\times9}(\mathbf{X}_r), \quad (10)$$

thereby capturing complementary local structures at different spatial scales. The resulting feature maps are concatenated along the channel dimension, followed by a nonlinear activation and a $1 \times 1$ projection to restore the original channel dimensionality. A residual connection is incorporated to preserve low-level information and facilitate gradient propagation:

$$\mathbf{X}_{\mathrm{loc}} = W_{\mathrm{proj}}\phi\left([\mathbf{P}_3\|\mathbf{P}_9]\right) + \mathbf{X}, \qquad (11)$$

where $\phi(\cdot)$ denotes an element-wise nonlinearity.

Finally, a lightweight feed-forward network (FFN) is applied to further refine the local representation,

$$\mathbf{P}_{\mathrm{loc}} = \mathrm{FFN}(\mathbf{X}_{\mathrm{loc}}) + \mathbf{X}_{\mathrm{loc}}. \qquad (12)$$

This processing pathway parameterizes the local sufficiency component of the learned regularizer, denoted by $\mathcal{R}_{\mathrm{local}}(\mathbf{X})$. Specifically, smaller convolutional kernels are effective in modeling sharp mask-induced discontinuities and pixel-wise modulation artifacts, whereas larger kernels enhance robustness to short-range spectral displacement effects. The use of depth-wise convolutions preserves spatial stationarity while avoiding the premature oversmoothing often associated with early-stage global attention mechanisms.

**Latent Window Interaction Module for Local and Global Scale Bridging:** While multi-kernel token mixing enforces pixel-level sufficiency aligned with the coded

aperture, resolving the spectral-spatial entanglement introduced by dispersion $\mathcal{D}$ requires interaction beyond fixed local neighborhoods. To this end, we introduce the Latent Window Interaction Module (LWIM), illustrated in Fig. 2(b), which enables efficient cross-region reasoning by operating in a compact latent token space.

Given an input feature map $\mathbf{X} \in \mathbb{R}^{C \times H \times W}$, LWIM first aggregates the spatial domain into a latent grid $\mathbf{X}_{\mathrm{win}} \in \mathbb{R}^{C \times \frac{H}{l} \times \frac{W}{l}}$ via mean pooling with a kernel size of $l \times l$ (e.g., $l = 4$). Each latent token summarizes local regional statistics while suppressing high-frequency noise and mask-induced artifacts. After projecting $\mathbf{X}_{\mathrm{win}}$ into a latent embedding space through a point-wise convolution, we apply a window-based Transformer block to the latent grid. To maintain a strict computational budget, this grid is partitioned into latent windows, and self-attention is performed within each:

$$\mathbf{Z}_1 = \mathrm{W}-\mathrm{MSA}(W_{\mathrm{proj}}\,\mathbf{X}_{\mathrm{win}}) + W_{\mathrm{proj}}\,\mathbf{X}_{\mathrm{win}}, \qquad (13)$$

Crucially, because each token in $\mathbf{X}\mathrm{win}$ represents an $l \times l$ patch of the original resolution, a window in the latent space covers a significantly larger physical area of the feature. This hierarchical partitioning allows for efficient long-range interaction, effectively capturing early-stage global contextual dependencies at a fraction of the computational cost of full-resolution attention. Subsequently, a feed-forward network further refines the latent representation,

$$\mathbf{Z}_2 = \mathrm{FFN}(\mathbf{Z}_1) + \mathbf{Z}_1, \qquad (14)$$

after which the latent features are bilinearly upsampled and projected back to the dense feature space,

$$\mathbf{P}_{\mathrm{lat}} = W_{\mathrm{restore}}(\uparrow \mathbf{Z}_2) \in \mathbb{R}^{C \times H \times W}. \qquad (15)$$

This restoration step injects spatially-broad contextual information into the original resolution without overwhelming the localized structures dictated by the coded aperture.

Finally, the outputs of the local MK-PTM and latent LWIM pathways are fused via channel-wise concatenation followed by a $1 \times 1$ convolution and a residual connection,

$$\mathbf{X}_{\mathrm{out}} = W_{\mathrm{fuse}}[\,\mathbf{P}_{\mathrm{loc}}\,\|\,\mathbf{P}_{\mathrm{lat}}\,] + \mathbf{X} \qquad (16)$$

ensuring stable gradient flow during early unfolding iterations.

Together, MK-PTM and LWIM decompose the learned regularizer as

$$\mathcal{R}(\mathbf{X}) = \mathcal{R}_{\mathrm{local}}(\mathbf{X}) + \mathcal{R}_{\mathrm{latent}}(\mathbf{X}), \qquad (17)$$

explicitly resolving the local-global dissonance induced by the CASSI forward operator, serving as the first stage of the

*Table 1.* Comparisons with the SoTA methods on 10 simulation scenes. Top row: PSNR (dB); bottom row: SSIM.

| Methods | GFLOPs | s1 | s2 | s3 | s4 | s5 | s6 | s7 | s8 | s9 | s10 | Avg |
|---|---|---|---|---|---|---|---|---|---|---|---|---|
| | | E2E methods | | | | | | | | | | |
| TSA-Net (Meng et al., 2020b) | 110.06 | 32.03 | 31.00 | 32.25 | 39.19 | 29.39 | 31.44 | 30.32 | 29.35 | 30.01 | 29.59 | 31.46 |
| | | 0.892 | 0.858 | 0.915 | 0.953 | 0.884 | 0.908 | 0.878 | 0.888 | 0.890 | 0.874 | 0.894 |
| HDNet (Hu et al., 2022) | 154.76 | 35.14 | 35.67 | 36.03 | 42.30 | 32.69 | 34.46 | 33.67 | 32.48 | 34.89 | 32.38 | 34.97 |
| | | 0.935 | 0.940 | 0.943 | 0.969 | 0.946 | 0.952 | 0.926 | 0.941 | 0.942 | 0.937 | 0.943 |
| MST-S (Cai et al., 2022b) | 12.96 | 34.71 | 34.45 | 35.32 | 41.50 | 31.90 | 33.85 | 32.69 | 31.69 | 34.67 | 31.82 | 34.26 |
| | | 0.930 | 0.925 | 0.943 | 0.967 | 0.933 | 0.943 | 0.911 | 0.933 | 0.939 | 0.926 | 0.935 |
| MST-M (Cai et al., 2022b) | 18.07 | 35.15 | 35.19 | 36.26 | 42.48 | 32.49 | 34.28 | 33.29 | 32.40 | 35.35 | 32.53 | 34.94 |
| | | 0.937 | 0.935 | 0.950 | 0.973 | 0.943 | 0.948 | 0.921 | 0.943 | 0.942 | 0.935 | 0.943 |
| MST-L (Cai et al., 2022b) | 28.15 | 35.40 | 35.87 | 36.51 | 42.27 | 32.77 | 34.80 | 33.66 | 32.67 | 35.39 | 32.50 | 35.18 |
| | | 0.941 | 0.944 | 0.953 | 0.973 | 0.947 | 0.955 | 0.925 | 0.948 | 0.949 | 0.941 | 0.948 |
| CST-S (Cai et al., 2022a) | 11.67 | 34.78 | 34.81 | 35.42 | 41.84 | 32.29 | 34.49 | 33.47 | 32.89 | 34.96 | 32.14 | 34.71 |
| | | 0.930 | 0.931 | 0.944 | 0.967 | 0.939 | 0.949 | 0.922 | 0.945 | 0.944 | 0.932 | 0.940 |
| CST-M (Cai et al., 2022a) | 16.91 | 35.16 | 35.60 | 36.57 | 42.29 | 32.82 | 35.15 | 33.85 | 33.52 | 35.28 | 32.84 | 35.31 |
| | | 0.938 | 0.942 | 0.953 | 0.972 | 0.948 | 0.956 | 0.927 | 0.952 | 0.946 | 0.940 | 0.947 |
| CST-L (Cai et al., 2022a) | 27.81 | 35.82 | 36.54 | 37.39 | 42.28 | 33.40 | 35.52 | 34.44 | 33.83 | 35.92 | 33.36 | 35.85 |
| | | 0.947 | 0.952 | 0.959 | 0.972 | 0.953 | 0.962 | 0.937 | 0.961 | 0.951 | 0.948 | 0.954 |
| CST-L+ (Cai et al., 2022a) | 40.01 | 35.96 | 36.84 | 38.16 | 42.44 | 33.25 | 35.72 | 34.86 | 34.34 | 36.51 | 33.09 | 36.12 |
| | | 0.949 | 0.955 | 0.962 | 0.975 | 0.955 | 0.963 | 0.944 | 0.961 | 0.957 | 0.945 | 0.957 |
| $S^2$-Tran (Wang et al., 2025) | 27.21 | 36.17 | 37.57 | 37.29 | 42.96 | 34.40 | **36.44** | 35.41 | 34.50 | 36.54 | 33.57 | 36.48 |
| | | 0.949 | 0.958 | 0.957 | 0.975 | 0.960 | 0.965 | 0.946 | 0.963 | 0.959 | 0.952 | 0.958 |
| DWMT (Luo et al., 2024) | 46.71 | 36.46 | 37.75 | 38.47 | 44.23 | 33.99 | 36.17 | 35.22 | 34.56 | 37.41 | **34.00** | 36.82 |
| | | 0.957 | **0.963** | 0.965 | 0.984 | 0.963 | **0.970** | 0.949 | **0.968** | 0.965 | **0.959** | 0.964 |
| **HSRA-S (Our)** | 9.01 | 35.85 | 36.92 | 37.45 | 43.93 | 34.15 | 35.06 | 34.97 | 33.57 | 36.30 | 32.60 | 36.08 |
| | | 0.947 | 0.949 | 0.955 | 0.978 | 0.956 | 0.955 | 0.941 | 0.951 | 0.947 | 0.938 | 0.952 |
| **HSRA-M (Our)** | 13.05 | 36.46 | 37.49 | 38.64 | 44.63 | 34.46 | 35.71 | 35.92 | 34.07 | 37.24 | 33.11 | 36.77 |
| | | 0.954 | 0.957 | 0.961 | 0.983 | 0.961 | 0.963 | 0.953 | 0.959 | 0.958 | 0.947 | 0.960 |
| **HSRA-L (Our)** | 27.34 | **36.90** | **38.12** | **39.69** | **45.38** | **35.34** | 36.23 | **36.32** | **34.82** | **38.09** | **34.00** | **37.49** |
| | | **0.958** | 0.962 | **0.968** | **0.986** | **0.967** | 0.969 | **0.956** | 0.966 | 0.964 | 0.957 | **0.965** |
| | | Deep Unfolding Models (DUMs) | | | | | | | | | | |
| DGSMP (Huang et al., 2021) | 85.77 | 33.26 | 32.09 | 33.06 | 40.54 | 28.86 | 33.08 | 30.74 | 31.55 | 31.66 | 31.44 | 32.63 |
| | | 0.915 | 0.898 | 0.925 | 0.964 | 0.882 | 0.937 | 0.886 | 0.923 | 0.911 | 0.925 | 0.917 |
| GAP-Net (Meng et al., 2020a) | 78.58 | 33.74 | 33.26 | 34.28 | 41.03 | 31.44 | 32.40 | 32.27 | 30.46 | 33.51 | 30.24 | 33.26 |
| | | 0.911 | 0.900 | 0.929 | 0.967 | 0.919 | 0.925 | 0.902 | 0.905 | 0.915 | 0.895 | 0.917 |
| ADMM-Net (Ma et al., 2019) | 78.58 | 34.12 | 33.62 | 35.04 | 41.15 | 31.82 | 32.54 | 32.42 | 30.74 | 33.75 | 30.68 | 33.58 |
| | | 0.918 | 0.902 | 0.931 | 0.966 | 0.922 | 0.924 | 0.896 | 0.907 | 0.915 | 0.895 | 0.918 |
| DAUHST-3stg (Cai et al., 2022c) | 27.17 | 36.56 | 37.92 | 39.36 | 44.97 | 34.82 | 36.22 | 35.99 | 34.24 | 38.50 | 33.63 | 37.22 |
| | | 0.953 | 0.960 | 0.968 | 0.985 | 0.964 | 0.968 | 0.953 | 0.963 | 0.967 | 0.954 | 0.963 |
| PADUT-3stg (Li et al., 2023) | 22.91 | 36.16 | 37.83 | 39.55 | 44.43 | 34.56 | 35.56 | 35.61 | 33.70 | 38.14 | 33.18 | 36.87 |
| | | 0.953 | 0.963 | 0.971 | 0.985 | 0.965 | 0.967 | 0.951 | 0.963 | 0.965 | 0.950 | 0.963 |
| RDLUF-3stg (Dong et al., 2023) | 38.45 | 36.67 | 38.48 | 40.63 | 46.04 | 34.63 | 36.18 | 35.85 | 34.37 | 38.98 | 33.73 | 37.56 |
| | | 0.953 | 0.965 | 0.971 | 0.986 | 0.963 | 0.966 | 0.951 | 0.963 | 0.966 | 0.950 | 0.963 |
| SSR-3stg (Zhang et al., 2024b) | 28.38 | **38.11** | 39.91 | 42.33 | 46.86 | 36.42 | 37.23 | 37.49 | 35.59 | 41.20 | 34.99 | 39.01 |
| | | 0.967 | 0.975 | 0.979 | 0.991 | 0.974 | 0.975 | 0.967 | 0.973 | 0.979 | 0.965 | 0.974 |
| **HSRA-DUM-3stg (Our)** | 37.39 | 37.85 | 40.85 | 42.82 | 47.21 | 37.01 | 37.69 | 38.30 | 36.07 | 40.96 | 35.25 | 39.40 |
| | | **0.970** | 0.981 | 0.981 | 0.992 | 0.978 | 0.980 | 0.971 | 0.977 | 0.979 | 0.969 | 0.978 |
| **HSRA-DUM-4stg (Our)** | 49.82 | 37.81 | **41.25** | **43.40** | **47.83** | **37.51** | **38.13** | **38.72** | **36.50** | **41.48** | **35.44** | **39.81** |
| | | **0.970** | **0.983** | **0.982** | **0.993** | **0.980** | **0.981** | **0.974** | **0.979** | **0.981** | **0.970** | **0.979** |

encoder before deeper hierarchical attention enforces full global consistency.

**Hierarchical Multi-Granularity Spatially-Shifted Attention:** To enforce global structural coherence while preserving spatial fidelity, we introduce Hierarchical Multi-Granularity Spatially-Shifted Attention (HM-SSA), as illustrated in Fig. 2(c). HM-SSA is designed as a degradation-aware, hierarchical attention operator that progressively expands the receptive field across the U-Net hierarchy, explicitly mirroring the global integration behavior of the CASSI forward operator $\mathcal{S} \circ \mathcal{D}$.

Given an input feature map $\mathbf{X}$, HM-SSA first applies Physics-Guided Feature Reweighting (PGFR) to condition feature responses using the coded aperture prior. Specif-

ically, the coded mask $\mathcal{M}$ is embedded into $\mathbf{M}_{\mathrm{emb}}$ via a $3 \times 3$ Conv followed by a dual-path block: a parallel residual branch ($1 \times 1$ Conv) and an attention branch ($3 \times 3$ depthwise Conv and Sigmoid activation) that modulates feature activations through element-wise multiplication. This operation enforces that subsequent attention mechanisms operate over physically admissible signal interactions. The motivation stems from the fact that the coded aperture $\mathcal{M}$ induces spatially varying observability in the measurement process; applying global attention directly on unconditioned features may otherwise introduce spurious long-range correlations that are inconsistent with the sensing geometry. The PGFR operation is formulated as

$$\mathbf{X}^{(0)} = \mathrm{Proj}(\mathbf{X} \odot \mathbf{M}_{\mathrm{emb}}) + \mathbf{X}, \qquad (18)$$

where the residual connection preserves original feature statistics while injecting physics-consistent reweighting. By aligning feature importance with the physical modulation pattern, PGFR provides a stable and meaningful initialization for hierarchical attention.

The core of HM-SSA consists of three Window Transformer Blocks (WTB) operating at multiple spatial granularities. In the first two blocks, self-attention is computed within local $w \times w$ windows to preserve fine-scale spatial coherence, while spatial shifting is alternated between successive blocks (i.e., non-shifted and half-window shifted configurations). This shifted-window strategy enables effective information exchange across window boundaries without incurring the computational cost of global attention. In contrast, the final block adopts a larger attention window (e.g., size $2w$ with zero shift), allowing the model to capture coarser spectral–spatial dependencies and long-range structural correlations. Formally, let $\mathbf{X}^{(l)}$ denote the input to the $l$-th hierarchical attention block. The self-attention update is expressed as

$$\mathbf{X}_{\mathrm{SA}}^{(l)} = \mathrm{WTB}(\mathbf{X}^{(l)}) + \mathbf{X}^{(l)}, \qquad (19)$$

followed by a feed-forward refinement with residual stabilization,

$$\mathbf{X}^{(l+1)} = \mathrm{FFN}(\mathbf{X}_{\mathrm{SA}}^{(l)}) + \mathbf{X}_{\mathrm{SA}}^{(l)}. \qquad (20)$$

Through this progressive expansion of attention granularity, HM-SSA enables a smooth transition from local spatial coherence to global structural alignment. Overall, HM-SSA parameterizes the *global coherence* component of the learned regularizer. By jointly integrating degradation-aware feature conditioning, shifted-window self-attention, and hierarchical granularity expansion, HM-SSA systematically resolves long-range spectral–spatial coupling while respecting local mask geometry. As a result, global reasoning emerges progressively and remains tightly consistent with the underlying physical imaging process.

### 3.3. Integration into Deep Unfolding

In addition to serving as an E2E reconstruction, HSRA is embedded into a deep unfolding framework as a physically grounded learned prior. Based on the regularized inverse problem in Eq. (5) and the HQS scheme in Eq. (6), each $\mathbf{Z}$-update in Eq. (7) is parameterized by a single HSRA module, yielding stage-wise refinement that explicitly respects the CASSI forward model. The learned prior is explicitly decomposed across scales: MK-PTM enforces local sufficiency aligned with the coded aperture $\mathcal{M}$, LWIM addresses dispersion-induced coupling from $\mathcal{D}$, and HM-SSA enforces global structural coherence consistent with $\mathcal{S} \circ \mathcal{D}$. This hierarchical design directly resolves the local–global dissonance identified in Section 3.2. Embedding HSRA into the unfolding dynamics systematically suppresses global error propagation caused by the back-projection $\mathcal{A}^{\top}(\mathcal{A}(\mathbf{X}) - \mathbf{Y})$. The conceptual pipeline of the DUM implementation is illustrated in the upper part of Fig. 1. Compared to purely end-to-end models, the unfolded formulation offers improved stability under severe undersampling and noise, while maintaining clear interpretability through its optimization correspondence.

## 4. Experiments

We evaluate the proposed HSRA on both simulated and real HS datasets to assess reconstruction accuracy and computational efficiency. The evaluation protocol follows prior established practices (Meng et al., 2020b; Cai et al., 2022b).
**Datasets:** Experiments are conducted on the CAVE (Yasuma et al., 2010) and KAIST (Choi et al., 2017) datasets. CAVE contains 32 indoor scenes with a spatial resolution of $512 \times 512$ and is used for training, while ten KAIST scenes (up to $2704 \times 3376$) are reserved for testing. All scenes are spectrally truncated to 28 bands within 450–650 nm. Real compressive measurements are acquired using a physical CASSI system (Meng et al., 2020b), enabling evaluation under practical imaging conditions.

**Implementation Details:** HS images are cropped into $256 \times 256$ (simulated) and $660 \times 660$ (real) patches. The CASSI process is simulated with a random coded aperture and dispersion step $d = 2$, yielding measurements of size $256 \times 310$ and $660 \times 714$. Data augmentation includes random flips and rotations. All models are implemented in PyTorch and trained on an NVIDIA RTX A6000 using the Adam optimizer with RMSE loss. Unless otherwise stated, the channel reduction ratio $r$ in MK-PTM is set to 1, the latent window size in LWIM is fixed to 4, and HM-SSA employs window sizes of 8 and 16 to model non-local dependencies.

**Comparison with SoTA methods:** HSRA is compared with representative E2E and deep unfolding methods on the KAIST test set. Results are summarized in Table 1. Follow-

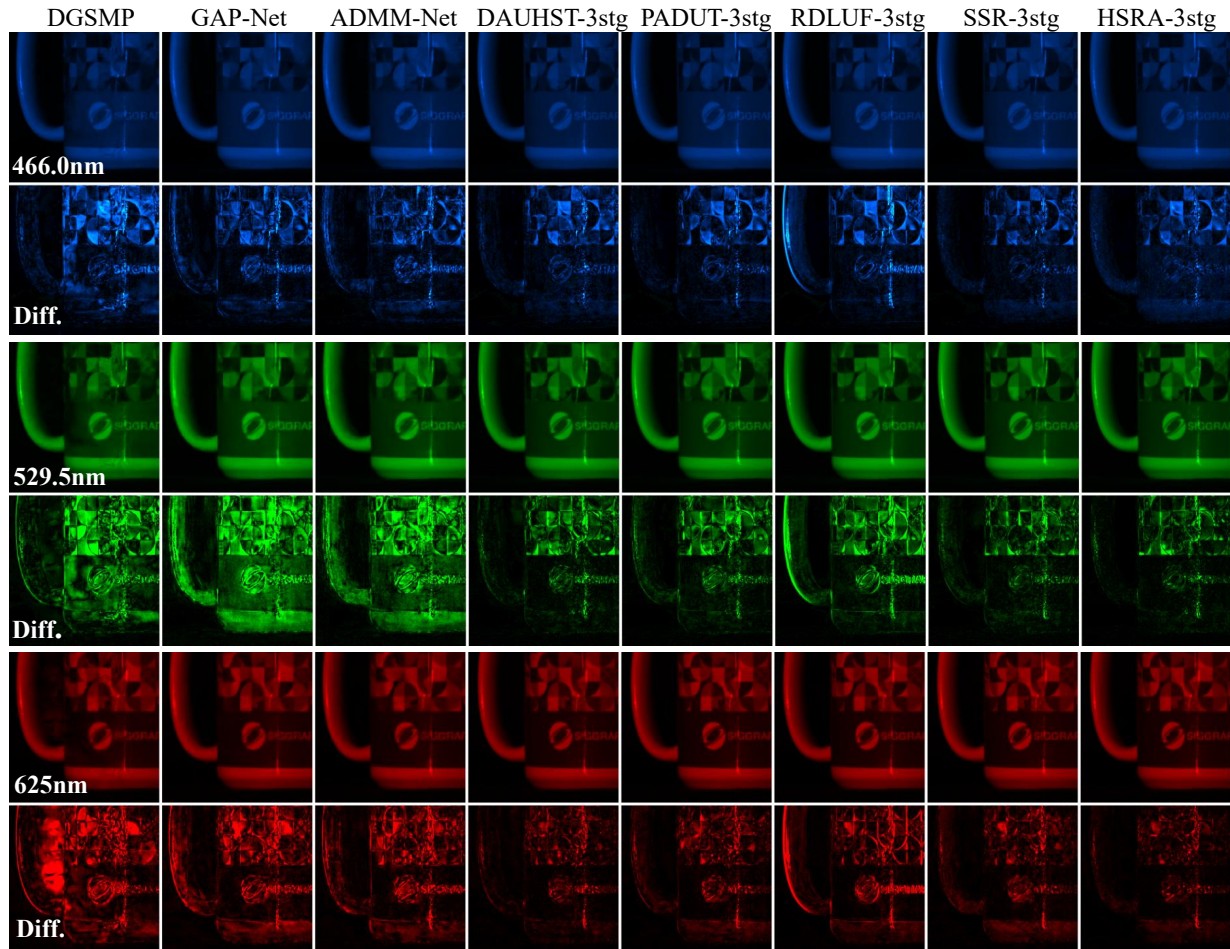

*Figure 3.* Visual comparison on a representative simulated scene, showing 3 spectral channels (out of 28) and the corresponding difference maps between the reconstructed images and the ground truth.

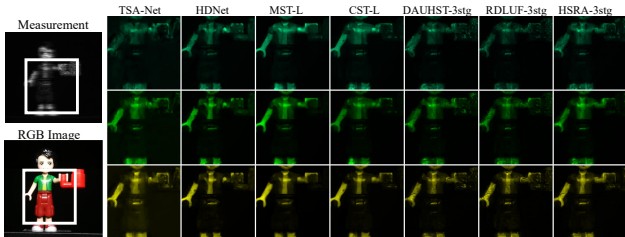

*Figure 4.* Visual comparison on a real-world scene, showing 3 spectral channels (out of 28).

ing (Cai et al., 2022b;a), three HSRA variants are evaluated: HSRA-S (28–28–28 channels), HSRA-M (28–56–56), and HSRA-L (56–56–56). HSRA-L achieves the best overall performance with an average PSNR/SSIM of 37.49 dB / 0.965, outperforming strong transformer-based baselines such as CST-L+, $S^2$-Tran, and DWMT. Notably, HSRA-L requires only 27.34 GFLOPs, approximately half the computational cost of DWMT (46.7), highlighting the favorable accuracy–efficiency trade-off enabled by our HSRA.

We evaluate our prior within DUMs across 3 and 4 stages.

HSRA-DUM-3stg achieves 39.40dB/0.978, outperforming the previous best SSR-3stg (39.01dB/0.974). Notably, HSRA-DUM-4stg reaches the highest overall performance of 39.81dB/0.982, validating our module's robust reconstruction and superior convergence behavior. Overall, both the E2E and DUM variants of HSRA consistently outperform existing SoTA approaches in accuracy–efficiency trade-offs, highlighting the effectiveness of the proposed stability and non-local modeling mechanism in HS image reconstruction. Perceptual quality is further examined in Fig. 3, where RGB visualizations and corresponding difference maps confirm that HSRA produces smaller reconstruction errors. Furthermore, Qualitative results on real HS measurements (Fig. 4) show that HSRA consistently preserves fine-grained details and spatial coherence, demonstrating strong robustness and generalization in real-world scenarios.

**Ablation Study:** We conduct a systematic ablation study on HSRA-S to isolate the contribution of each architectural component in resolving the proposed Local-Global Dissonance. As summarized in Table 2, the ablations are

*Table 2.* Ablation study on HSRA-S. Modules are grouped according to their role in resolving local–global dissonance. Besides global reconstruction quality (PSNR/SSIM), we further report scale-specific metrics: SAM for local spectral fidelity, LPSNR(8/16/32) for mid-scale structural consistency, and ERGAS for global spatial-spectral coherence.

| Category | Configuration | GFLOPs | PSNR↑ | SSIM↑ | SAM↓ | LPSNR(8)↑ | LPSNR(16)↑ | ERGAS↓ |
|---|---|---|---|---|---|---|---|---|
| Homogeneous Token Mixing | All-ConvFormer | 8.78 | 35.40 | 0.944 | 8.110 | 40.995 | 40.132 | 23.591 |
| | All-MK-PTM | 8.41 | 35.47 | 0.944 | 8.361 | 40.862 | 40.028 | 23.258 |
| | All-SwinFormer ($8\times8$) | 8.80 | 34.98 | 0.941 | 7.666 | 41.079 | 40.137 | 24.209 |
| | All-HM-SSA | 9.47 | 35.11 | 0.943 | 7.712 | 4.103 | 40.213 | 24.123 |
| Local Sufficiency | MK-PTM, $r$=0.5 | 8.71 | 35.90 | 0.950 | 7.780 | 41.297 | 40.447 | 22.106 |
| | MK-PTM, $3\times3$ only | 8.64 | 35.92 | 0.950 | 8.077 | 41.246 | 40.404 | 21.930 |
| | $r$=1, kernels $\{3, 9\}$ | – | $\geq$ +0.16 | +0.002 | ↓ | ↑ | ↑ | ↓ |
| Mid-Scale Interaction | W/o LWIM | 8.23 | 35.89 | 0.948 | 8.112 | 41.142 | 40.303 | 22.068 |
| | With LWIM (ours) | – | +0.19 | +0.004 | ↓ | ↑ | ↑ | ↓ |
| Global Coherence | W/o large-window SA ($2w$) | 6.64 | 35.65 | 0.945 | 8.304 | 40.890 | 40.040 | 22.631 |
| | Full HM-SSA | – | +0.43 | +0.007 | ↓ | ↑ | ↑ | ↓ |
| **Full Model** | **HSRA-S (Ours)** | **9.01** | **36.08** | **0.952** | **7.684** | **41.530** | **40.661** | **21.542** |

organized according to the functional role of each module across local sufficiency, mid-scale interaction, and global coherence. Besides conventional reconstruction metrics (PSNR/SSIM), we further introduce scale-specific evaluation metrics to quantitatively analyze representation behavior across different regimes. Specifically, SAM measures local spectral fidelity at the pixel level, reflecting the preservation of intrinsic spectral signatures under spatial-spectral entanglement. LPSNR(8/16) evaluates mid-scale structural consistency over different local regions, characterizing the capability of each model to maintain coherent structures across spatial scales. ERGAS evaluates global spatial-spectral coherence, measuring the overall consistency of reconstructed HS structures. These metrics enable a more interpretable analysis of how each architectural component contributes to reconciling local and global constraints.

First, homogeneous token-mixing baselines, where all blocks adopt a single mechanism (ConvFormer, MK-PTM, Swin Transformer, or HM-SSA), consistently underperform the proposed HSRA design across both global and scale-specific metrics. In particular, homogeneous architectures exhibit inferior SAM and ERGAS scores, indicating that single-form token mixing cannot simultaneously preserve local spectral fidelity and global structural coherence. This confirms that no single token-mixing strategy is sufficient to address both mask-induced local constraints and global spectral-spatial coupling. Second, ablations on the local sufficiency module demonstrate that MK-PTM is critical for stabilizing early-stage reconstruction. Reducing channel capacity ($r = 0.5$) or restricting MK-PTM to a single kernel degrades both SAM and LPSNR performance, while employing multi-kernel perception $(3, 9)$ consistently improves local spectral fidelity and mid-scale structural consistency. These results validate the necessity of scale-diverse local modeling for mitigating spectral distortion and recovering fine-grained structures. Third, removing the LWIM leads

to a noticeable drop in reconstruction quality, accompanied by consistent degradation in LPSNR metrics. This indicates that mid-scale interaction is essential for compensating dispersion-induced spectral-spatial entanglement and resolving boundary inconsistencies prior to global reasoning. Finally, disabling the large-window attention in HM-SSA significantly degrades PSNR, SSIM, and ERGAS performance despite reduced computational cost, highlighting the importance of hierarchical multi-granularity attention for enforcing long-range structural coherence. The simultaneous degradation in SAM and ERGAS further suggests that global reasoning also contributes to stabilizing local spectral representations through cross-scale interaction.

Overall, the full HSRA-S model achieves the best trade-off between accuracy and efficiency across all evaluation regimes. The consistent improvements in SAM, LPSNR, and ERGAS demonstrate that progressively reconciling local, mid-scale, and global constraints is essential for robust hyperspectral reconstruction.

## 5. Conclusions

This study introduced a physics-consistent Hierarchical Scale-Reconciling Architecture (HSRA) to address the Local–Global Dissonance inherent in compressive HS imaging. By aligning the learned prior with the spatial–spectral entanglement and non-commutative structure of the CASSI forward model, HSRA reconciles locally reliable observations with globally coherent spectral–spatial representations through hierarchical multi-scale interaction. Integrated into a deep unfolding framework, the proposed model serves as a scale-aware reconciliation operator rather than a generic denoiser, enabling stable optimization under severe undersampling and noisy conditions. Extensive experiments demonstrate that the proposed method consistently achieves superior reconstruction accuracy, robustness, spectral fidelity, and computational efficiency compared with SoTA methods.

## Acknowledgements

This work was supported in part by JSPS KAKENHI Grant-in-Aid for Scientific Research (B), Grant Number 26K02987.

## Impact Statement

This paper presents work whose goal is to advance the field of machine learning, specifically in the domain of computational imaging and hyperspectral reconstruction. There are many potential societal consequences of our work, none of which we feel must be specifically highlighted here.

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

# A. Theoretical Foundation: Scale-Conditional Identifiability

## A.1. Preliminaries

This appendix provides a detailed and rigorous exposition of the theoretical foundations underlying the proposed scale-conditional identifiability framework. The goal is not to establish the mathematical uniqueness of HS recovery under CASSI, which is impossible due to inherent rank deficiency, but rather to clarify how a scale-ordered learned prior stabilizes reconstruction and controls ambiguity propagation in a deep-unfolded optimization scheme.

## A.2. Forward Model and Structural Sources of Ambiguity

The CASSI forward operator is defined as:

$$\mathcal{A} = \mathcal{S} \circ \mathcal{D} \circ \mathcal{M}, \tag{21}$$

where we identify three distinct physical constraints that necessitate a hierarchical prior:

- $\mathcal{M}$ (Spatial Modulation): Performs spatial masking via a coded aperture. It introduces *local spatial undersampling*, where information is lost at pixel locations where the mask is zero.

- $\mathcal{D}$ (Dispersion/Shearing): Applies wavelength-dependent spatial shifts. Critically, $\mathcal{D}$ is a **bijective and invertible** operator; it does not destroy information but redistributes it spatially, coupling distant pixels across different wavelengths.

- $\mathcal{S}$ (Spectral Integration): Sums the dispersed cube along the spectral dimension into a 2D measurement. This operator is **rank-deficient** and is the primary source of the global null space $\mathcal{N}(\mathcal{A})$.

These three effects imply that $\mathcal{A}$ is neither block-diagonal nor scale-separable. Ambiguities arising at fine spatial scales (due to $\mathcal{M}$) can be propagated and amplified by the global spectral integration ($\mathcal{S}$) if not controlled.

## A.3. HQS Reconstruction and Error Decomposition

The Half-Quadratic Splitting (HQS) iteration alternates between a data-consistency update:

$$\mathbf{X}^{(k+1)} = \arg \min_{\mathbf{X}} \frac{1}{2} \|\mathcal{A}(\mathbf{X}) - \mathbf{Y}\|_2^2 + \frac{\rho}{2} \|\mathbf{X} - \mathbf{Z}^{(k)}\|_2^2, \tag{22}$$

and a prior-driven update:

$$\mathbf{Z}^{(k+1)} = \mathrm{Prox}_{\rho^{-1}\mathcal{R}}(\mathbf{X}^{(k+1)}). \tag{23}$$

Let $\mathbf{E}^{(k)} = \mathbf{X}^{(k)} - \mathbf{X}^\star$ denote the reconstruction error relative to a feasible solution $\mathbf{X}^\star$. We decompose the error as:

$$\mathbf{E}^{(k)} = \mathbf{E}_{\parallel}^{(k)} + \mathbf{E}_{\perp}^{(k)}, \tag{24}$$

where $\mathbf{E}_{\parallel}^{(k)} \in \mathcal{N}(\mathcal{A})$ and $\mathbf{E}_{\perp}^{(k)} \in \mathcal{N}(\mathcal{A})^{\perp}$. Since the data-consistency term penalizes only $\mathbf{E}_{\perp}^{(k)}$, components in the null space $\mathcal{N}(\mathcal{A})$ may persist or grow without scale-aware regularization.

## A.4. Local Mask Admissibility and Fine-Scale Stability

**Assumption A.1** (Local Mask Admissibility). For any spatial patch $\mathbf{X}_p$, there exists a non-empty subset of pixels within the patch where the coded aperture is non-zero, such that:

$$\|\mathcal{M}(\mathbf{X}_p)\|_2^2 \geq \alpha \|\mathbf{X}_p\|_2^2, \qquad \alpha > 0. \tag{25}$$

**Lemma A.2** (Patch-Level Energy Preservation). *For any perturbation $\Delta_p$ intersecting the non-zero support of the mask:*

$$\|\mathcal{M}(\Delta_p)\|_2 \geq \sqrt{\alpha}\, \|\Delta_p\|_2. \tag{26}$$

This establishes conditional local identifiability: fine-scale ambiguities are energetically constrained where measurements are physically present.

## A.5. Spectral Integration and the Global Null Space

The summation operator $\mathcal{S}$ maps $B$ spectral channels to a single measurement, inducing a null space:

$$\mathcal{N}(\mathcal{A}) = \left\{ \mathbf{U} : \sum_{\lambda=1}^{B} (\mathcal{D} \circ \mathcal{M})(\mathbf{U})_\lambda = 0 \right\}. \tag{27}$$

Because $\mathcal{D}$ redistributes spectral components spatially, null-space elements exhibit long-range structured correlations, motivating the need for mid- and global-scale constraints.

## A.6. Scale-Decomposed Regularization and Ordered Suppression

We assume the regularizer $\mathcal{R}$ decomposes as $\mathcal{R} = \mathcal{R}_{\mathrm{loc}} + \mathcal{R}_{\mathrm{mid}} + \mathcal{R}_{\mathrm{glob}}$.

**Theorem A.3** (Scale-Conditional Stability). *Let the forward operator $\mathcal{A}$ satisfy Assumption A.1. If the prior is applied in increasing order of spatial scale (Local $\to$ Mid $\to$ Global), then:*

1. *The reconstruction error $\mathbf{E}^{(k)}$ remains bounded.*

2. *The residual null-space components $\|\mathbf{E}_\|^{(k)}\|_2$ are monotonically suppressed across iterations.*

*Sketch.* Local regularization (MK-PTM) suppresses high-frequency perturbations. Mid-scale regularization (LWIM) aligns spatially shifted structures across wavelengths, resolving dependencies introduced by $\mathcal{D}$. Global regularization (HM-SSA) constrains the remaining low-frequency ambiguities in $\mathcal{N}(\mathcal{A})$. This hierarchy prevents fine-scale errors from being amplified by the integration operator $\mathcal{S}$. □

## A.7. Stability of the Plug-and-Play Scheme

**Assumption A.4** (Lipschitz Denoiser). The learned prior $f_{\boldsymbol{\theta}}$ is $L$-Lipschitz continuous with $L \leq 1$.

Under this assumption, the prior update is non-expansive. Combined with the strong convexity of the data-consistency subproblem, standard PnP theory guarantees that the HQS iteration converges to a fixed point $\hat{\mathbf{X}}$.

## A.8. Interpretation and Architectural Implications

This analysis establishes that scale-aligned, hierarchical priors are essential in CASSI. Specifically: (i) MK-PTM enforces stability under masking; (ii) LWIM resolves dispersion-induced ambiguities; and (iii) HM-SSA constrains the spectral-collapse null space.

# B. Reconstruction Robustness to Measurement Noise

To systematically evaluate robustness under realistic sensing perturbations, we conduct noise sensitivity experiments by injecting additive white Gaussian noise (AWGN) into the compressive measurements. We consider a range of noise levels defined by the standard deviation $\sigma \in \{0.0001, 0.0005, 0.0010, 0.0015\}$, applied to input measurements normalized to a dynamic range of $[0, 1]$. Reconstruction performance is assessed using both pixel-wise fidelity (PSNR) and structural similarity (SSIM). In addition to absolute metrics, we report the relative performance degradation (%) with respect to the noise-free case ($\sigma = 0$), which provides a clearer characterization of each model's sensitivity to measurement noise. The compared results are shown in Table 2

**Absolute Reconstruction Performance:** From Table 3, it is observed that the proposed HSRA variants consistently outperform the homogeneous baseline (ALL-HM-SSA) in both PSNR and SSIM across all noise levels. In the noise-free setting ($\sigma$=0), HSRA-L achieves the highest reconstruction quality, confirming that hierarchical scale reconciliation improves representation fidelity. While all methods exhibit a monotonic decrease in performance as $\sigma$ increases—reflecting the ill-posed nature of CASSI reconstruction—the HSRA variants maintain a significant performance margin. Notably, HSRA-L preserves the highest absolute PSNR and SSIM even at the maximum noise level ($\sigma$=0.0015).

**Relative Degradation and Noise Sensitivity:** A more informative comparison emerges when examining relative performance drops. For PSNR, the homogeneous ALL-HM-SSA baseline shows a pronounced sensitivity to noise, with

*Table 3.* Reconstruction performance under increasing measurement noise. Noise levels represent the standard deviation ($\sigma$) of additive Gaussian noise applied to measurements with a dynamic range of $[0, 1]$.

| Noise ($\sigma$) | PSNR (dB) | | | | PSNR Drop (%) | | | |
|---|---|---|---|---|---|---|---|---|
| | ALL-HM-SSA | HSRA-S | HSRA-M | HSRA-L | ALL-HM-SSA | HSRA-S | HSRA-M | HSRA-L |
| 0 | 35.112 | 36.079 | 36.772 | 37.489 | – | – | – | – |
| 0.0001 | 35.053 | 36.079 | 36.772 | 37.488 | 0.171 | 0.000 | 0.000 | 0.001 |
| 0.0005 | 35.050 | 36.077 | 36.768 | 37.481 | 0.177 | 0.007 | 0.010 | 0.020 |
| 0.0010 | 35.041 | 36.067 | 36.748 | 37.450 | 0.205 | 0.034 | 0.066 | 0.103 |
| 0.0015 | 35.028 | 36.042 | 36.725 | 37.400 | 0.239 | 0.105 | 0.127 | 0.236 |
| | SSIM | | | | SSIM Drop (%) | | | |
| | ALL-HM-SSA | HSRA-S | HSRA-M | HSRA-L | ALL-HM-SSA | HSRA-S | HSRA-M | HSRA-L |
| 0 | 0.9435 | 0.9517 | 0.9597 | 0.9653 | – | – | – | – |
| 0.0001 | 0.9425 | 0.9517 | 0.9597 | 0.9653 | 0.106 | 0.000 | 0.000 | 0.001 |
| 0.0005 | 0.9424 | 0.9516 | 0.9596 | 0.9651 | 0.120 | 0.005 | 0.009 | 0.021 |
| 0.0010 | 0.9421 | 0.9514 | 0.9592 | 0.9644 | 0.146 | 0.029 | 0.049 | 0.085 |
| 0.0015 | 0.9417 | 0.9508 | 0.9587 | 0.9637 | 0.193 | 0.086 | 0.098 | 0.160 |

degradation increasing rapidly from 0.17% at noise $= 0.0001$ to 0.24% at noise $\sigma = 0.0015$. This behavior indicates that globally enforced attention across all stages tends to amplify noise-corrupted features, particularly when local identifiability is weak. In contrast, HSRA-S and HSRA-M exhibit substantially improved robustness. Under low noise levels ($\sigma \leq 0.0005$), their PSNR drops remain negligible (below 0.01%), and degradation increases smoothly as noise intensifies. Although HSRA-L shows larger relative drops at higher noise levels, this is largely attributable to its higher baseline accuracy; importantly, its absolute reconstruction quality remains superior across all noise regimes.

**Structural Robustness (SSIM):** Similar trends are observed for SSIM, which is more sensitive to structural distortions. ALL-HM-SSA exhibits steadily increasing SSIM drops from 0.11% to 0.19% as noise increases, indicating strong vulnerability to noise-induced structural artifacts. In contrast, HSRA-S shows virtually no SSIM degradation under low noise and remains below 0.01% at noise $\sigma = 0.0005$, while HSRA-M demonstrates similarly stable behavior. Although HSRA-L experiences larger relative SSIM drops at higher noise levels, it consistently achieves the highest absolute SSIM, indicating superior preservation of spatial–spectral structure.

**Architectural Interpretation:** These results suggest that robustness is not solely determined by model capacity, but critically depends on how physical constraints are reconciled across scales. The homogeneous ALL-HM-SSA architecture enforces global interactions uniformly, causing measurement noise to be rapidly propagated through long-range dependencies. In contrast, HSRA mitigates noise amplification through its hierarchical design: (i) local sufficiency: Enforced in early stages (MK-PTM) to prevent noise from contaminating global representations; (ii) Progressive Receptive Fields: Receptive fields expand gradually (LWIM), allowing global coherence to be imposed only after local ambiguities are stabilized; and (iii) Physical Alignment: Learned priors are aligned with the physical sensing process, resulting in graceful degradation.

Overall, the PSNR and SSIM analyses confirm that HSRA achieves a superior noise-accuracy trade-off. Beyond improving reconstruction fidelity under clean measurements, HSRA exhibits significantly more graceful degradation under increasing noise, which is essential for practical CASSI systems operating under non-ideal sensing conditions.

## C. Additional Visualization Results

Due to space constraints in the main paper, the qualitative results were limited to visual comparisons between the proposed deep-unfolding implementation with HSRA-based prior learning and existing deep-unfolding methods (DUM). Consequently, qualitative comparisons between the proposed end-to-end (E2E) HS reconstruction network, HSRA, and recent state-of-the-art E2E approaches were not included.

To provide a more complete qualitative evaluation, this supplemental material presents three additional visualization examples:

- Two representative examples comparing E2E-based hyperspectral reconstruction methods, including the proposed HSRA and several state-of-the-art E2E networks.

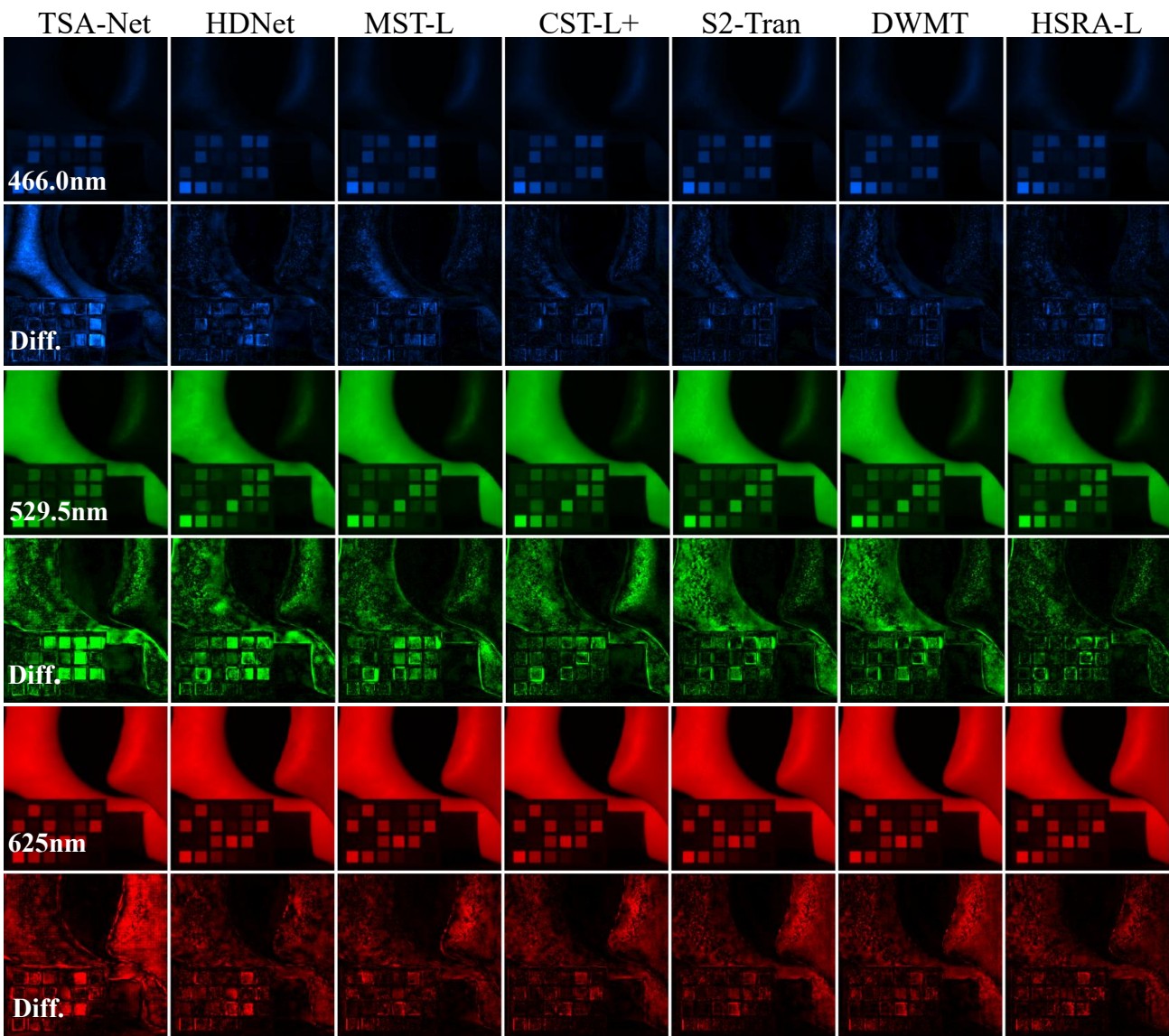

*Figure 5.* Qualitative comparison under end-to-end reconstruction on Scene 3. Results of the proposed HSRA and representative state-of-the-art E2E reconstruction networks are shown for three selected spectral bands (out of 28), together with the corresponding difference maps relative to the ground truth.

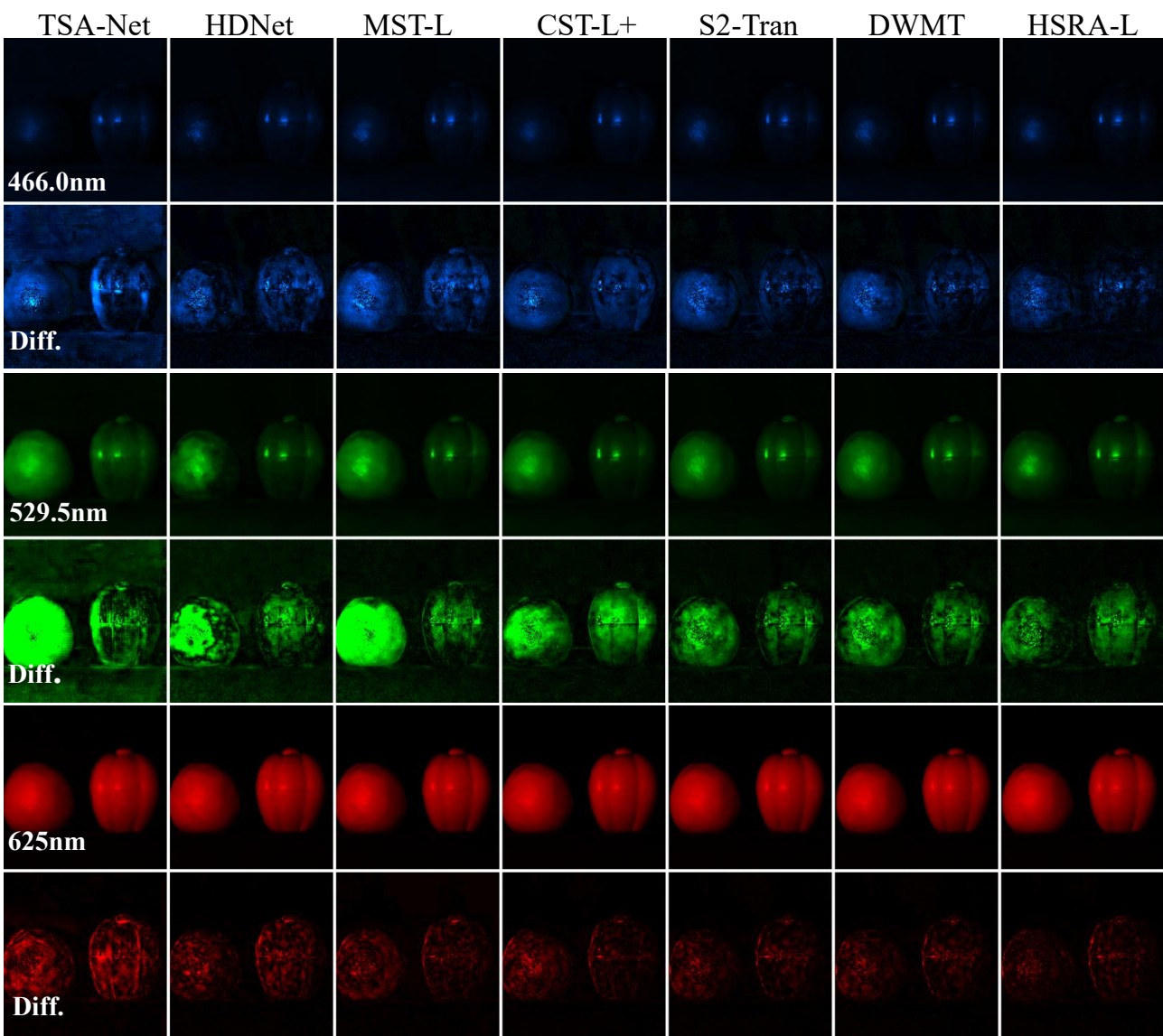

*Figure 6.* Qualitative comparison under end-to-end reconstruction on Scene 4. Results of the proposed HSRA and representative state-of-the-art E2E reconstruction networks are shown for three selected spectral bands (out of 28), together with the corresponding difference maps relative to the ground truth.

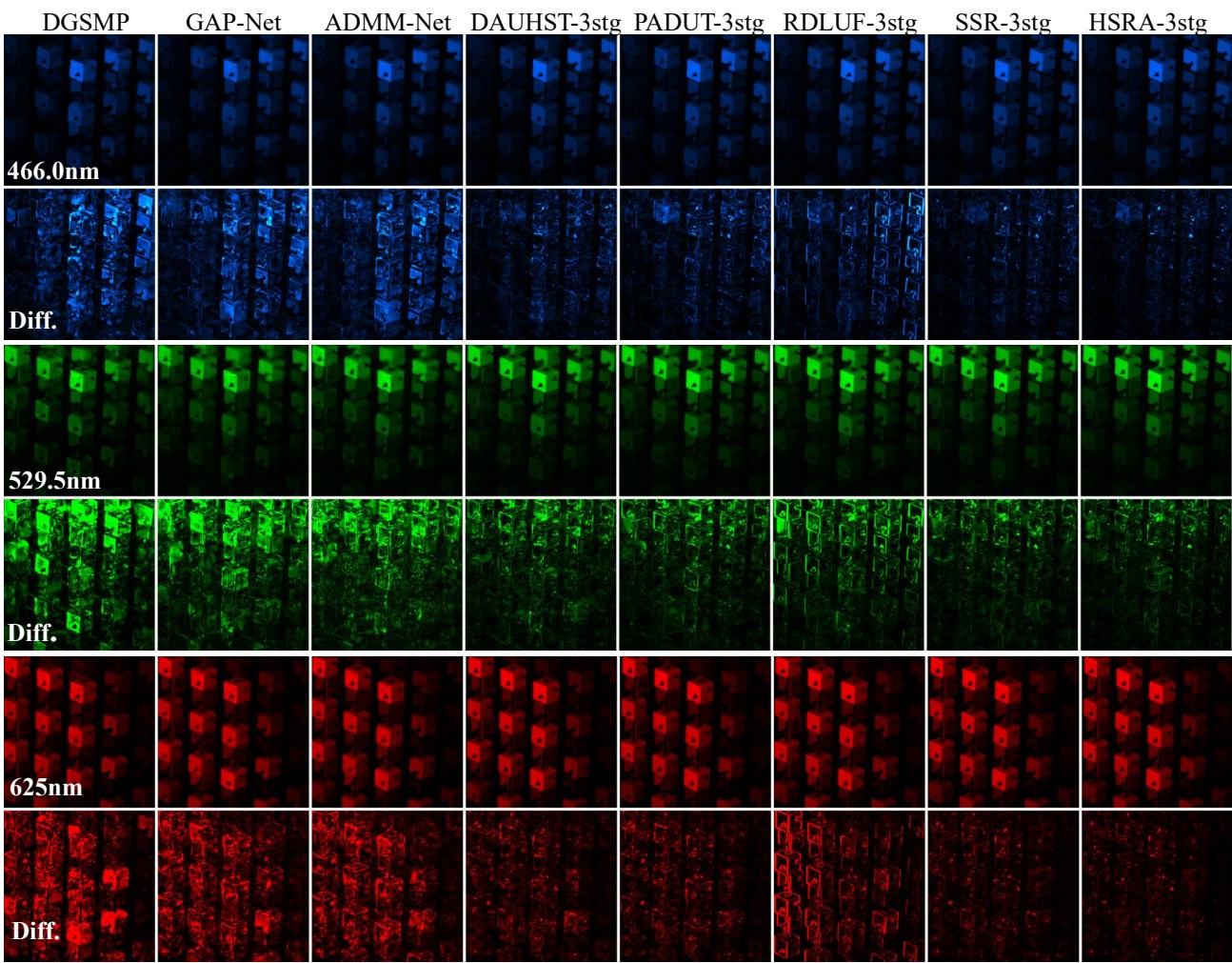

*Figure 7.* Additional qualitative comparison under the deep-unfolding framework on Scene 2. Reconstruction results obtained by the proposed deep-unfolding implementation with HSRA-based prior learning are compared with representative DUM-based approaches. Three representative spectral channels are selected from the 28-band hyperspectral data. For each method, the reconstructed images and the corresponding absolute difference maps with respect to the ground truth are presented to visually evaluate reconstruction accuracy and error characteristics.

- One additional example comparing DUM-based reconstruction methods, complementing the limited qualitative results shown in the main paper.

**Visualization Comparison of E2E Methods.** Figures 5 and 6 present qualitative reconstruction results on two representative HS scenes using various end-to-end learning–based methods. For each scene, we show:

1. Reconstructed hyperspectral images produced by different E2E approaches, including HSRA-L, TSA-Net (Meng et al., 2020b), MST-L (Cai et al., 2022b), CST-L+ (Cai et al., 2022a), $S^2$-Tran (Wang et al., 2025), and DWMT (Luo et al., 2024);

2. The corresponding absolute difference maps with respect to the ground truth, visualizing reconstruction errors.

As illustrated in these figures, the proposed HSRA consistently yields reconstructions with sharper spatial structures and improved spectral consistency, particularly in regions containing fine textures and abrupt spectral changes.

**Visualization Comparison of DUM Methods.** Figure 7 provides an additional qualitative comparison between the proposed deep-unfolding framework with HSRA-based prior learning and several representative DUM-based baselines. This example highlights the benefit of incorporating hierarchical contextual modeling and transformer-based reconstruction into the unfolding framework, resulting in more accurate detail recovery and fewer artifacts compared to conventional unfolding methods.

