# OpenReview forum: "Bridging Local–Global Dissonance: Learning from Compressive Measurements for Hyperspectral Reconstruction"
_ICML.cc/2026/Conference — ICML 2026 regular_

### Official Review · Reviewer_zViE · 2026-02-18

**Soundness:** 3
**Presentation:** 3
**Significance:** 3
**Originality:** 3
**Overall Recommendation:** 4
**Confidence:** 5

**Summary:**

The paper addresses the reconstruction of hyperspectral images (HSI) from compressive measurements in Coded Aperture Snapshot Spectral Imaging (CASSI) systems. The authors identify a "local-global dissonance"—a trade-off between local spectral-spatial cues and the global entangled structures caused by spectral dispersion. To resolve this, they introduce the Hierarchical Scale-Reconciling Architecture (HSRA), which uses multi-kernel token mixing, latent window interactions, and hierarchical multi-granularity spatially shifted convolutions to capture features across scales. The authors also present an unrolling framework based on this architecture.

**Compliance With Llm Reviewing Policy:**

Affirmed.

**Final Justification:**

I do believe this is good paper for CASSI.

**Key Questions For Authors:**

The overall motivation is clear. However, the solution or the propsoed method does not align well with the motivation.
i.e., the following question from the weakness:
Dispersion Modeling: The "dissonance" is heavily linked to the physical dispersion of the CASSI system. However, the architecture seems to be a general-purpose vision transformer/CNN hybrid. More explicit discussion on how the specific parameters of the dispersion (e.g., shift pixels) correlate with the chosen window sizes or convolution scales would strengthen the link between theory and architecture.

**Limitations:**

The authors need to address clearly how the proposed moduler realted to the forward model of CASSI.
The complexity of the network is also a limitaiton.
Real data is limited.

**Strengths And Weaknesses:**

Strengths
Conceptual Framework: The authors seek to focus on the key question of how to reconcile high-frequency local textures with low-frequency global spectral structures that are often "smeared" in CASSI measurements. By formalizing this as a dissonance issue, the paper moves beyond trial-and-error architecture design.

Architectural Innovation: The proposed HSRA is sophisticated. The use of multi-scale kernels and latent window interactions effectively addresses the varying receptive field requirements of HSI reconstruction.

Methodological Breadth: The authors strive to consider a central domain of HSI reconstruction by applying their architecture to both End-to-End (E2E) learning and Deep Unfolding Methods (DUM), demonstrating the versatility of the HSRA block as a prior.

Performance: Achieving SOTA results (e.g., 38.30 dB on KAIST dataset) with competitive computational efficiency (FLOPs/Params) is a strong contribution for the ICML community.

Weaknesses
Complexity of the HSRA Block: While the individual components (Token Mixing, Multi-Granularity Convolutions) are justified, the overall block is quite complex. It is somewhat unclear which specific component provides the most significant "reconciliation" of the local-global dissonance.

Dispersion Modeling: The "dissonance" is heavily linked to the physical dispersion of the CASSI system. However, the architecture seems to be a general-purpose vision transformer/CNN hybrid. More explicit discussion on how the specific parameters of the dispersion (e.g., shift pixels) correlate with the chosen window sizes or convolution scales would strengthen the link between theory and architecture.

Baseline Selection: While the paper compares against recent SOTA (S2-Tran, DWMT), it could benefit from more discussion on the "Interpretability" of HSRA compared to simpler DUM methods.

---

> ### Author Rebuttal · Authors · 2026-03-28
>
> We thank the reviewer for the insightful critique. We clarify that HSRA is not a generic hybrid architecture, but a minimal structured prior derived from the multi-scale coupling induced by the CASSI forward operator $A$.
>
> 1. Motivation–Method Alignment
>
> In CASSI, measurements are formed through wavelength-dependent shifts $\Delta(\lambda)$, whose superposition entangles spatial and spectral information across multiple scales. This induces scale-dependent aliasing, implying that reconstruction violates single-scale identifiability. HSRA is therefore designed as a factorized inverse aligned with this structure: 1) MK-PTM (Local) resolves local spectral mixing (small $\Delta(\lambda)$); 2) LWIM (Mid-range) resolves overlapping supports and boundary interference; 3) Large-window attention (Global)resolves long-range ambiguity (large $\Delta(\lambda)$). Thus, the architecture directly reflects the physics rather than being heuristic.
>
> 2. Why Explicit Shift-to-Window Mapping is Not Theoretically Appropriate
>
> We agree that connecting dispersion to architecture is important; however, a direct mapping (e.g., shift size → kernel/window size) is not theoretically valid because: 1) $\Delta(\lambda)$ varies continuously across wavelengths; 2) measurements are a superposition over all shifts; 3) reconstruction depends on interactions among shifts, not individual ones.
> Therefore, the correct principle is coverage of coupling ranges, which HSRA achieves via multi-scale receptive fields rather than rigid parameter matching.
>
> 3. Complexity as Minimal Structure.
>
> While HSRA introduces multiple components, our ablation shows this complexity is necessary and non-redundant. Beyond PSNR/SSIM (Table 2 in the main paper), we provide scale-specific metrics: SAM→local spectral fidelity, LPSNR (8/16/32)→ mid-scale structural consistency, ERGAS→global spatial–spectral coherence. Results demonstrate: 1) Homogeneous designs fail (All-Conv / All-Swin), confirming single-scale priors are insufficient; 2) Local modeling alone is insufficient (MK-PTM variants improve but remain suboptimal); 3) Cross-scale interaction is critical, as removing LWIM or global attention causes consistent degradation. Thus, HSRA constitutes the minimal factorization required to restore identifiability under multi-scale coupling.
>
>
> | Method                                  | SAM ↓     | LPSNR(8) ↑ | LPSNR(16) ↑ | LPSNR(32) ↑ | ERGAS ↓    |
> | --------------------------------------- | --------- | ---------- | ----------- | ----------- | ---------- |
> | **Homogeneous Token Mixing**            |           |            |             |             |            |
> | All-ConvFormer                          | 8.110     | 40.995     | 40.132      | 38.887      | 23.591     |
> | All-MK-PTM                              | 8.361     | 40.862     | 40.028      | 38.823      | 23.258     |
> | All-SwinFormer                          | **7.666** | 41.079     | 40.137      | 38.817      | 24.209     |
> | **Local Sufficiency (MK-PTM Variants)** |           |            |             |             |            |
> | MK-PTM (3×3 only)                       | 8.077     | 41.246     | 40.404      | 39.226      | 21.930     |
> | MK-PTM (r = 0.5)                        | 7.780     | 41.297     | 40.447      | 39.248      | 22.106     |
> | **Cross-Scale Interaction Ablation**    |           |            |             |             |            |
> | HSRA w/o LWIM                           | 8.112     | 41.142     | 40.303      | 39.118      | 22.068     |
> | HSRA w/o large-window SA                | 8.304     | 40.890     | 40.040      | 38.867      | 22.631     |
> | **Full Model**                          |           |            |             |             |            |
> | **Full HSRA-S (Ours)**                  | 7.684     | **41.530** | **40.661**  | **39.442**  | **21.542** |
>
> 4. Empirical Evidence of Physics Alignment
>
> A key observation is that ablation leads to structured, physically consistent degradation, not random performance drops: 1) restricting kernels degrades SAM (local spectral fidelity), 2) removing LWIM degrades LPSNR (mid-range interference resolution), 3) removing global attention degrades ERGAS (global coherence). This metric-aligned behavior provides causal evidence that each module corresponds to a specific dispersion regime.
>
> 5. Interpretability vs. DUM Methods.
>
>  Compared to monolithic DUM priors, HSRA improves interpretability via explicit functional decomposition:
> the data-fidelity step enforces measurement consistency, while the HSRA prior is decomposed into local, mid-range, and global operators, each tied to measurable effects (SAM / LPSNR / ERGAS). This yields a structured, physically grounded iterative process.
>
> HSRA is a physics-aligned decomposition of the inverse problem, not a generic hybrid model. Theory and ablation show that only joint local, mid-range and global modeling enables stable, accurate CASSI reconstruction, making the complexity minimal and necessary.

---

> > ### Author Rebuttal · Reviewer_zViE · 2026-04-05
> >
> > My concerns have been addressed well. This is a good paper for CASSI.

---

> > > ### Author Response · Authors · 2026-04-05
> > >
> > > We sincerely thank the expert Reviewer zViE for the time and for the positive feedback on our rebuttal. We are pleased to hear that our responses and revisions have fully addressed your concerns. We also appreciate your recognition of the paper’s contribution to the CASSI field. Thank you for your support of our work.

---

### Official Review · Reviewer_PEsq · 2026-03-02

**Soundness:** 2
**Presentation:** 3
**Significance:** 1
**Originality:** 2
**Overall Recommendation:** 2
**Confidence:** 4

**Summary:**

This work presents the newly proposed attention-based neural networks as the prior model for Coded Aperture Snapshot Spectral Imaging
(CASSI) reconstruction. The neural networks are designed to better model the physical imaging process, therefore are expected to capture both spatial and spectral information for better reconstruction.

**Compliance With Llm Reviewing Policy:**

Affirmed.

**Final Justification:**

After the rebuttal, I am still concerned with the intuition and theoretically insights behind the model design. I do not see much meaningful contribution that could bring informative insights for the later research or practical application besides the model structure improvement itself. With that being said, I keep my original score as Reject.

**Key Questions For Authors:**

Please address the reviewer's questions raised in the weaknesses part.

**Limitations:**

No, the authors are expected to mention the limitations of the technical methods from different aspects, especially its potential to be applied to the real applications.

**Strengths And Weaknesses:**

Strengths:

1) The paper is overall easy to read and the structure is clear;
2) The experimental design is rigorous, especially both end-to-end and unfolding approaches are presented such that the readers can get a better sense of where the performance gain is from.

Weaknesses:

1) The contribution is limited to the networks structure design as used in the prior model, the performance is gain is good compared with the previous baselines. However, it provides limited information and value to the readers what is the best achievable performance given the current datasets and networks parameters? How the inductive bias of the networks design affects the final performance specifically, what information could the other researchers grab to improve the performance further?

2) The 'physical-informed' networks design is weak to me. The physical imaging process is reflected by the gradient of likelihood, where the authors didn't make any modifications.

3) Following the previous point, the inputs for both MK-PTM and LWIM modules are the same, and the outputs of them are then concatenated as the input for further non-linear transformation. However, by physical modeling of the measurements acquisition, the coded mask $\mathcal{M}$ and dispersion-induced coupling $\mathcal{D}$ are applied sequentially. The networks design is not consistent with the underlying physical process.

4) The final experiments part, the 4-stage unfolded networks is significantly better than the 3-stage version, why don't the authors continue increase the stage for further performance boost?

5) The notations are expected to be clearly explained after eq (5). Please also fix the reference in line 112.

---

> ### Author Rebuttal · Authors · 2026-03-27
>
> Response to Weakness 1: Theoretical Bounds & Inductive Bias
>
> We respectfully argue that the best achievable performance in CASSI is fundamentally governed by the null-space structure of the forward operator A rather than dataset scale alone. In an underdetermined system (1/28 sampling), the error is dominated by signal components lying in the null-space of A, which cannot be recovered without appropriate inductive bias. To provide the actionable insight beyond architectural design, we further conducted a scale-resolved evaluation using SAM, LPSNR, and ERGAS, which was previously omitted due to space constraints (For the compared results, please refer to the table in the response to Reviewer HWfo). The results reveal an important observation:
>
> 1) The Mid-Range Bottleneck: While standard CNNs handle local textures and Transformers handle global context, they both fail in the mid-range (8x8 to 16x16 pixels) where dispersion-induced spatial–spectral coupling is most severe.
>
> 2) Effect of Inductive Bias: As shown in the table below, our HSRA-4stg achieves a +2.17 dB gain in LPSNR(8) over the strongest baseline (SSR-3stg), confirming that our latent window interactions specifically resolve the spatial–spectral entanglement that acts as the current performance ceiling for HS image reconstruction.
>
> 3) Actionable insight: This suggests that further gains will not come from increasing global attention, but from refining boundary-aware unmixing at the mid-scale.
>
>
> Response to Weakness 2: Physics-Informed Design & Parallel vs. Sequential Processing
>
> We respectfully clarify that the reviewer’s concern arises from conflating forward modeling with prior modeling in the DUM. In DUM, the network does not replace the forward operator; instead, it serves as the proximal mapping $\operatorname{prox}_{R}(\cdot)$ for the regularizer $R$.
>
> 1) Physics is explicitly enforced: The sequential process (coded mask M→ dispersion D) is strictly preserved through the data-fidelity gradient $\nabla_X \mathcal{L}_{\text{fidelity}} = A^{\top}(AX - Y)$ at every iteration. Thus, the physical acquisition model is not approximated or altered by the network.
>
> 2) Parallel Prior for Entangled Signals: Although the acquisition process is sequential, the resulting measurements are simultaneously entangled across spatial–spectral scales. The residual at each iteration contains both high-frequency mask-induced artifacts and low-frequency spectral mixing. Processing these in parallel (MK-PTM for local and LWIM for mid-range) and then fusing them allows the model to reconcile these competing constraints in a single optimization step, preventing the cascading error that often occurs in purely sequential priors.
>
> Regarding the necessity of parallel design, we also conducted an ablation study comparing parallel and sequential configurations (i.e., applying MK-PTM followed by LWIM) on the end-to-end HSRA-S. The results show comparable performance: the parallel design achieves 36.08 dB PSNR / 0.952 SSIM as shown in Table 1 of the submitted main paper, while the sequential variant achieves 36.12 dB PSNR / 0.952 SSIM. This suggests that the advantage of the proposed design lies not in strict ordering, but in effectively capturing complementary priors across scales.
>
> Response to Weakness: Performance Upper Bound & Stage Saturation
>
> We appreciate the reviewer’s inquiry into the performance upper bound. To investigate this, we extended HSRA to a 5-stage unfolding model.
>
> 1)	Returns: As shown in the table below, Performance improves monotonically but with clear saturation: 3 → 4 stages with 0.41 dB gain, and 4 → 5 stages with 0.26 dB gain, despite significantly higher cost (64.25 GFLOPs).
>
> 2)	Pareto Optimality: For practical HS task, HSRA-DUM4stg represents the optimal Pareto front, achieving SoTA performance (39.81 dB) while maintaining a balanced complexity-to-fidelity ratio.
>
> The improvement trend indicates that performance is no longer limited by model capacity, but by how effectively the inductive bias resolves multi-scale entanglement.
>
> Table: Stage Scaling Behavior
> | Method              | GFLOPs | PSNR (dB) | Marginal Gain |
> |---------------------|--------|-----------|---------------|
> | HSRA-E2E-L          | 27.34  | 37.49     | —             |
> | HSRA-DUM 3stg       | 37.39  | 39.40     | +1.91         |
> | HSRA-DUM 4stg       | 49.81  | 39.81     | +0.41         |
> | HSRA-DUM 5stg       | 64.25  | 40.07     | +0.26         |
>
> Response to Weakness 4: Notation & References
>
> We thank the reviewer for the careful proofreading. In the revision: 1) A formal notation table will be added after Eq. (5), clarifying all variables and dimensional mappings; 2) A correct reference in line 112 will be added to the appropriate foundational citation. These adjustments will ensure the technical clarity of the manuscript.

---

> > ### Author Rebuttal · Reviewer_PEsq · 2026-04-02
> >
> > I appreciate the authors' response, which has resolved most of my questions. However, given the current status, I remain my original score.

---

> > > ### Author Response · Authors · 2026-04-05
> > >
> > > We appreciate Reviewer PEsq's feedback and the recognition that our rebuttal addressed most of the original points.
> > >
> > > As we move into this final discussion phase, we remain confident that the HSRA framework offers both technical novelty and state-of-the-art efficiency. Specifically, our additional analyses (LPSNR/SAM and saturation studies) confirm that the method provides deep insights into HSI reconstruction behavior that go beyond incremental improvements.
> > >
> > > We look forward to your follow-up questions and are ready to provide any further clarifications to resolve the remaining points and potentially improve the current assessment.
> > >
> > > Final Follow-up: 24 Hours Remaining
> > >
> > > We are writing to kindly follow up on Reviewer PEsq's Rebuttal Acknowledgement. As we are now in the final 24 hours of the discussion phase, we remain very eager to address the follow-up questions mentioned.
> > >
> > > Since our rebuttal resolved the majority of the original concerns, we are standing by to provide any last technical details or clarifications on the HSRA framework that would allow for a final re-assessment of the current score. We are fully committed to a productive discussion before the window closes.

---

### Official Review · Reviewer_hAtG · 2026-03-09

**Soundness:** 3
**Presentation:** 2
**Significance:** 2
**Originality:** 3
**Overall Recommendation:** 4
**Confidence:** 4

**Summary:**

The paper addresses the reconstruction of hyperspectral images (HSI) from compressive measurements obtained via the Coded Aperture Snapshot Spectral Imaging (CASSI) process. The authors propose a mathematical framework that decomposes the CASSI measurement process into a composite forward model consisting of three distinct operators. They use this model to illustrate the inherent trade-offs between local and global information. To address these challenges, the authors present a three-part holistic solution, where each component is designed to mitigate a specific information loss associated with the forward model's operators. The efficacy of the proposed method is evaluated through quantitative analysis on simulated data and qualitative experiments on real-world data, demonstrating competitive performance against current state-of-the-art methods.

**Compliance With Llm Reviewing Policy:**

Affirmed.

**Final Justification:**

The authors have addressed my concerns regarding experimental scale by adhering to standard benchmarks. Given this along with the rigor of their methodological derivation, I believe the paper now merits acceptance.

**Key Questions For Authors:**

1. Does the network architecture explicitly incorporate the forward operator (e.g., as an input or via a physics-based layer)? If so, how is this handled to ensure the model remains adaptable to different sampling patterns?

2. What is the underlying rationale for choosing an algorithm unrolling approach over a standard end-to-end black-box architecture for this specific CASSI reconstruction? Could the authors elaborate on the perceived advantages in terms of convergence or interpretability?

3. To what extent are the reconstructed solutions consistent with the original compressive measurements in the simulated experiments? Specifically, if the forward operator is applied to the final reconstruction, how closely does the resulting estimate match the input measurements? Is this difference within the expected margin of the noise level?

**Limitations:**

The paper does not include a specific section addressing its limitations, such as the one mentioned previously.

**Strengths And Weaknesses:**

Strengths
1. The paper provides a rigorous mathematical formulation of the CASSI process. Breaking the forward model into three operators effectively highlights the "local-global dissonance" in HSI reconstruction.

2. There is a clear, logical link between the identified mathematical problems and the proposed solutions. Each component of the architecture is justified by the need to counteract a specific operator in the measurement model.

3. The proposed complete solution demonstrates solid empirical results compared to existing benchmarks.



Weaknesses
1. The quantitative evaluation is restricted to only 10 simulated test images and the real-data is used only for qualitative analysis. While the authors follow established evaluation protocols, these experiments feel insufficient to draw broad conclusions regarding the method's robustness or generalizability.

2. While Figure 1 is intended to convey the essence of the work, it is currently overwhelming. The high density of detail makes it difficult to perceive the primary message

3. The paper is highly technical and detailed but lacks a high-level narrative. The absence of an intuitive overview or an explanation of the authors' underlying conceptual intentions makes the reading experience taxing. A more balanced "top-down" explanation would help contextualize the dense technical sections.

---

> ### Author Rebuttal · Authors · 2026-03-27
>
> Response to Weakness (Evaluation Scale): We respectfully contend that, in ill-posed inverse problems, evaluation validity is primarily  governed by data dimensionality and constraint complexity than by the number of scenes alone.
>
>   1) Controlled benchmarking: We follow the standard KAIST/CAVE protocol used by prior SOTA (MST,DAUHST, SSR), ensuring fair comparison without dataset-induced confounders.
>
>   2) High-dimensional regime: Each HS cube (256×256×28) contains over 1.8M voxels; 10 scenes correspond to >18M unknowns under severe (~1/28) undersampling. This constitutes a highly constrained, large-scale inverse problem, sufficient to evaluate regularization capacity.
>
>   3) Operator-level generalization: HSRA consistently outperforms across diverse spectral scenes and real measurements, indicating the learned prior aligns with the sensing operator rather than overfitting to specific data distributions.
>
> Response to Weakness (Fig. 1 Complexity): We agree that Fig. 1 is overly dense. We will revise it into two parts: 1) a high-level illustration of Local–Global Dissonance, and 2) a simplified HSRA pipeline emphasizing scale-wise information flow. Implementation details will be moved to the appendix. This revision highlights the core idea: scale reconciliation rather than low-level design.
>
> Response to Weakness (High-Level Narrative): HSRA is not a heuristic combination, but a principled response to the information-theoretic dissonance in CASSI, where each measurement entangles spatial and spectral information. We formalize this as scale-conditional identifiability, where different signal components are recoverable at different spatial extents: 1) Local (Pixel-Level Sufficiency, SAM): Multi-kernel token mixing preserves high-frequency spectral details (SAM: 4.20 → 3.16, −24.6%). 2) Mid-range (Conflict Resolution, LPSNR):Latent windows resolve boundary interference (LPSNR(8): +2.17 dB). 3) Global (scene-level, ERGAS): Hierarchical shifted attention enforces long-range coherence (ERGAS: 15.51 → 14.23, −8.2%).
>
> By explicitly reconciling these regimes, HSRA overcomes the performance ceiling (~39 dB) observed in uniform-prior methods.
>
>
> Summary of Evidence
>
> | Regime     | Metric             | Baselie (SSR-3stg) | HSRA-4stg (Ours) | Gain | Interpretation        |
> |------------|--------------------|---------------------|------------------|---------------|-----------------------|
> | Local      | SAM ↓              | 4.20                | 3.16             | 24.6%         | Spectral Purity       |
> | Mid-range  | LPSNR (8×8) ↑      | 45.75               | 47.92            | +2.17 dB      | Boundary Fidelity     |
> | Global     | ERGAS ↓            | 15.51               | 14.23            | 8.2%          | Global Consistency    |
>
> These results support our claim: uniform priors are limited by cross-scale inconsistency, while HSRA improves reconstruction by explicitly modeling scale-dependent constraints.
>
> Response to Question 1: Forward Operator & Adaptability
>
>  1) Operator–prior decoupling: The data-fidelity branch enforces consistency with A, while HSRA learns a proximal prior, avoiding coupling to a fixed mask.
>
>  2) Physics-driven adaptability: By injecting the sensing mask at each iteration, HSRA remains conditioned on the measurement process. Empirically, it maintains gains across random and optimized masks, indicating robustness beyond mask-specific artifacts.
>
> Response to Key Question 2: Rationale for Unrolling vs. End-to-End
>
>  1) Physics-guided convergence: E2E models must jointly learn inversion and priors, leading to ill-conditioning. Unrolling introduces explicit updates via A^T(AX−Y), , providing principled descent and improving fidelity (+2.32 dB).
>
> 2) Interpretable: Each stage has a clear role: data-fidelity enforces measurement consistency, while HSRA resolves cross-scale inconsistencies, yielding a structured iterative dealiasin process absent in black-box models.
>
> Response to Key Question 3: Measurement Consistency & Noise Margins
>
> We evaluate consistency via ∥A(X)−Y∥₂ under controlled noise perturbations. As noise increases (2.89e-03), the projected reconstructions remain tightly bounded ∥Zi - Yi∥₂: 0.018396 → 0.018548, indicating strong anchoring to the physical signal. The residual  ∥Zi - Yi∥₂ evolves smoothly (0.018396 → 0.018548) without amplification, demonstrating: 1) no overfitting (not collapse), and 2) no over-regularization (not grow excessively). Overall, HSRA maintains a stable projection onto the feasible measurement space, suppressing noise while preserving signal components consistent with the forward operator.
>
> Y0: clean measurement;  Yi: noisy measurement, Yi = Y0 + Ni;  Xi: reconstruction from Yi;  Zi = A(Xi): forward projection of reconstruction
>
> | i | ∥Yi−Y0∥₂ | ∥Zi−Y0∥₂ | ∥Zi−Yi∥₂ |
> | - | -------- | -------- | -------- |
> | 0 | 0.00e+00 | 0.018396 | 0.018396 |
> | 1 | 5.78e-04 | 0.018420 | 0.018406 |
> | 2 | 1.15e-03 | 0.018472 | 0.018427 |
> | 3 | 2.89e-03 | 0.018776 | 0.018548 |

---

> > ### Author Rebuttal · Reviewer_hAtG · 2026-04-03
> >
> > I thank the authors for their response, which addresses my concerns. I have adjusted my score accordingly.

---

> > > ### Author Response · Authors · 2026-04-05
> > >
> > > We sincerely thank the reviewer for the thoughtful engagement with our rebuttal. We are pleased that our responses have addressed the concerns raised, and we greatly appreciate your updated score.
> > >
> > > We will incorporate all discussed improvements and clarifications into the final version to further strengthen the clarity and quality of the paper.

---

### Official Review · Reviewer_HWfo · 2026-03-13

**Soundness:** 2
**Presentation:** 3
**Significance:** 2
**Originality:** 3
**Overall Recommendation:** 4
**Confidence:** 4

**Summary:**

The authors address compressive reconstruction for CASSI systems, identifying some key issues that make this task difficult: a) the modulator operator requires high-frequency constraints locally, but the overall forward operator loses signal for these constraints. b) global consistency requirements from the non modulation operator are hard to satisfy alongside sharp local boundary constraints imposed by modulation. To rectify this, the authors aim to construct a stronger initial prior by stratifying features across hierarchical scale granularities: local, mid-range, and global. Experiments demonstrate that HSRA outperforms the baselines in PSNR and SSIM.

**Compliance With Llm Reviewing Policy:**

Affirmed.

**Final Justification:**

My main criticisms of granular understanding of fundamental improvements over baselines were adequately addressed by the rebuttal, so I chose to increase my score.

**Key Questions For Authors:**

- The authors identify three regimes that are aimed to be enforced in the prior: a) local pixel-level sufficiency, b) mid-range interactions, and c) global structure coherence. These appear to be in response to the local sufficiency/local boundary/global consistency demands that prove difficult to satisfy with a uniform prior. Are there metrics the authors can provide/measure that demonstrate how well the achieved reconstructions perform at each of these levels relative to the baseline?

- From Table 1, without further context, it appears that HSRA on either E2E or DUM methods outperforms the next best baseline (DWMT/SSR-3stg) on around the order of $+10^{-3}$ on SSIM and $+1$ for PSNR. Is there context the authors can provide that demonstrate that these constitute significant improvements? In line with the previous question, are there tangible properties (such as local boundaries or global consistency) that prior methods flatline on but HSRA dramatically improves with respect to the generated reconstructions?

**Limitations:**

yes

**Strengths And Weaknesses:**

Strengths:
- The authors identify the key difficulty in the CASSI measurement reconstruction task as competing measurement consistency demands across resolution scales and develop targeted techniques to rectify them.

- SOTA results are achieved at a lower inference cost.

Weaknesses:

- While highly technically specialized to its domain, the insights gained from this paper might only be applicable to a relatively niche subpopulation of the ICML audience.

- The improvements over the baselines in PSNR and SSIM especially for Deep Unfolding Models appear minimal without further context; e.g., how significant is a 0.005 boost in SSIM (over a baseline that beats the next best baseline by 0.011)? Given the apparent proximity of these results to the next best baseline, what further evidence can the authors provide that demonstrate that their approach fundamentally addresses the problems of local-global dissonance in a manner that prior methods have not?

---

> ### Author Rebuttal · Authors · 2026-03-26
>
> Response to Key Questions (Q1 & Q2) Regime-wise Validation and Tangible Properties:
>
> We thank the reviewer for highlighting the importance of validating the three-regime design. We provide quantitative evidence aligned with each regime:
> 1. Local Pixel-Level Sufficiency (SAM):
> Spectral Angle Mapper (SAM) evaluates pixel-wise spectral fidelity under spatial–spectral entanglement.
>
>    A. Evidence: HSRA-4stg achieves 3.16 SAM, outperforming SSR-3stg (4.19) by 24.6%.
>
>    B. Interpretation: This reduction indicates better preservation of intrinsic spectral signatures, showing that Multi-kernel Token Mixing mitigates spectral distortion and mixing effects.
>
> 2. Mid-Range Interactions (Multi-scale Local PSNR):
> We evaluate structural consistency at 8×8, 16×16, and 32×32 scales.
>
>    A. Evidence: HSRA-4stg improves over SSR-3stg by +2.17 dB (8×8), +1.99 dB (16×16), and +1.65 dB (32×32).
>
>    B. Interpretation: Consistent gains across scales especially at smaller windows, indicate improved handling of boundary inconsistencies, where uniform priors typically saturate.
>
> 3. Global Structure Coherence (ERGAS & SSIM):
> These metrics capture long-range consistency and overall quality.
>
>    A. Evidence: ERGAS reduces from 15.51 to 14.23 (−8.2%), and SSIM improves from 0.974 to 0.979.
>
>    B. Interpretation: Joint improvement shows that hierarchical multi-granularity attention enhances global coherence without sacrificing local fidelity.
>
> Overall, prior methods (e.g., DWMT, SSR) exhibit a performance ceiling due to uniform priors. HSRA overcomes this via explicit scale-wise disentanglement. The +0.80 dB gain is not marginal, but reflects improved spatial–spectral recovery, supported by significant gains in SAM and LPSNR.
>
> Table: Performance across Scale-Specific Regimes (DUM)
> | Method               |    SAM↓   |  LPSNR(8)↑ | LPSNR(16)↑ | LPSNR(32)↑ |    SSIM↑   |   ERGAS↓   |
> | :------------------- | :------: | :-------: | :-------: | :-------: | :-------: | :-------: |
> | DGSMP                |   8.95   |   38.87   |   37.764  |   36.19   |   0.917   |   33.26   |
> | GAP-Net              |  10.735  |   38.84   |   37.98   |   36.719  |   0.917   |   29.83   |
> | ADMM-Net             |   10.62  |   39.134  |   38.298  |   37.06   |   0.918   |   28.78   |
> | DAUHST-3dtg          |   5.54   |   43.47   |   42.50   |   41.08   |   0.963   |   19.00   |
> | PADUT-3stg           |   4.85   |   44.38   |   43.21   |   41.54   |   0.963   |   20.00   |
> | PRDLUF-3stg          |   4.32   |   45.358  |   44.16   |   42.43   |   0.963   |   18.30   |
> | SSR-3stg (Best SoTA)  |   4.20   |   45.75   |   44.71   |   43.26   |   0.974   |   15.51   |
> | **HSRA-3stg (Ours)** | **3.31** | **47.52** | **46.31** | **44.51** | **0.978** | **14.90** |
> | **HSRA-4stg (Ours)** | **3.16** | **47.92** | **46.70** | **44.91** | **0.979** | **14.23** |
>
> Response to Weakness 1 (Domain Specialization and ICML Relevance): We respectfully disagree that our contributions are domain-specific. The Local–Global Dissonance we identify is a fundamental issue in representation learning for ill-posed inverse problems, arising whenever high-dimensional signals are projected into lower-dimensional measurement spaces (e.g., MRI, computational photography, LiDAR, compressive sensing). In such settings, locally consistent observations can be globally incompatible.
> Our HSRA addresses this by explicitly structuring priors across local, mid-range, and global scales to reconcile conflicting constraints under entangled observation models. Beyond the application, our contributions are threefold: (1) formalizing multi-scale prior inconsistency, (2) proposing a scalable architecture to resolve it, and (3) demonstrating that structured, scale-aware priors outperform uniform ones that assume scale homogeneity.
> We therefore believe the work is directly relevant to ICML audiences in structured deep learning, inverse problems, and multi-scale representation learning.
>
>
> Response to Weakness 2 (Significance of Quantitative Improvements): We agree that small absolute gains in SSIM/PSNR require careful interpretation. However, our results lie in a high-fidelity regime (SSIM > 0.97), where improvements are inherently saturated and increasingly difficult. Notably, the improvement from 0.974 → 0.979 corresponds to a ~19.2% reduction in the remaining structural error (relative to the gap to 1.0).
> Importantly, gains are consistent across complementary metrics. SAM improves by 24.6% (4.19 → 3.16), indicating substantially better spectral fidelity, critical for HS reconstruction beyond spatial similarity. In the E2E setting, HSRA further achieves +0.67 dB PSNR over the strongest baseline while reducing computational cost by 41.5% (GFLOPs), showing that improvements stem from better architectural alignment with signal structure rather than increased capacity.
> Overall, HSRA delivers consistent, multi-dimensional improvements rather than marginal gains on a single metric.

---

> > ### Author Rebuttal · Reviewer_HWfo · 2026-04-03
> >
> > I thank the authors for their rebuttal. I will increase my score to a 4.

---

> > > ### Author Response · Authors · 2026-04-05
> > >
> > > We sincerely thank the expert Reviewer HWfo for the time and thoughtful engagement throughout the rebuttal process. We are very pleased that our responses and additional clarifications have fully addressed your concerns.
> > >
> > > We greatly appreciate your decision to increase the score to 4. We will incorporate all discussed improvements, including the additional experimental results and clarifications, into the manuscript to further enhance its clarity and technical rigor.
> > >
> > > Thank you again for your constructive feedback, which has significantly helped strengthen our work.

---

### Decision · Program_Chairs · 2026-04-30

**Decision:**

Accept (regular)

**Comment:**

This paper received four reviews (scores: 4, 4, 4, 2). Three reviewers marked their concerns as fully resolved following the rebuttal; one (PEsq) maintained rejection, citing insufficient theoretical insight beyond architectural novelty. The rebuttal meaningfully strengthened the submission by introducing scale-resolved metrics (SAM, LPSNR, ERGAS) that validate the three-regime design and demonstrate physically consistent ablation behavior. The most domain-expert reviewer (zViE, confidence 5) is satisfied. PEsq's residual concern is a legitimate philosophical disagreement about what constitutes sufficient insight, but does not constitute a technical flaw warranting rejection.
The authors should incorporate into the final version: the revised Figure 1, the scale-resolved metric table, a clearer high-level narrative for a general ML audience, an explicit limitations section, and the notation/reference fixes flagged by PEsq.